ꝺ | **Open Peer Review** | Clinical Microbiology | Research Article

# Dynamics of the respiratory infectome in children with community-acquired pneumonia: insights from large and short time-scale analyses

Yuqi Liao,[1,2] Jinxia Cheng,[1] Suwan Xiong,[3,4] Yanshan Liu,[5] Jun Qian,[4] Mang Shi,[1,2] Yun Guo,[4] Yan-Jun Kang[5]

**ABSTRACT**  Community-acquired pneumonia (CAP) has emerged as a significant health challenge for young children, especially after the relaxation of COVID-19 restrictions, which coincided with a sharp increase in CAP cases. While pathogen profiling is commonly performed, comprehensive studies examining the total infectome and its dynamic changes during disease progression and in relation to the pandemic remain scarce. To address this gap, we conducted a prospective cohort study involving 58 children hospitalized with CAP in Wuxi, China, during and after COVID-19 control measures. Sputum samples were analyzed using metagenomic and metatranscriptomic sequencing to characterize the total infectome. Results showed that RNA sequencing offers a more comprehensive view of the infectome, while DNA sequencing excels in detecting DNA viruses with greater sensitivity. Notable increases in *Mycoplasma pneumoniae*, human respiratory syncytial virus (RSV), and *Haemophilus influenzae* were observed after COVID-19 restrictions were lifted. During disease progression, some patients exhibited a decline in pathogen abundance, while others developed secondary infections, frequently involving co-infections, which might contribute to prolonged pneumonia or complicated disease course. Viral-bacterial co-infections were common, with *M. pneumoniae* and RSV being the most prevalent combination. In summary, this study highlights the shifting respiratory infectome in children with CAP, both after the relaxation of COVID-19 control measures and throughout hospitalization. It emphasizes the need for comprehensive infectome monitoring to track dynamic changes across broader timeframes and during disease progression, offering insights for improved clinical management and future research.

**IMPORTANCE**  Community-acquired pneumonia (CAP) remains a leading threat to children's health globally, with shifting pathogen dynamics post-COVID-19 posing new challenges. This study reveals how pandemic control measures and their relaxation influenced the respiratory "infectome"—the full spectrum of pathogens—in children with CAP. By integrating multi-sequencing technologies, we uncovered critical trends: a resurgence of virulent pathogens like *Mycoplasma pneumoniae* and respiratory syncytial virus after restrictions eased, frequent viral-bacterial co-infections linked to prolonged pneumonia, and distinct infection patterns during hospitalization that predict recovery or complications. These findings highlight the need for dynamic, multi-pathogen surveillance to guide clinical decisions, particularly in managing co-infections and preventing secondary infections. Our work provides actionable insights for pediatricians and public health experts to anticipate post-pandemic pathogen behavior, tailor treatments, and mitigate risks during future outbreaks, ultimately improving care for vulnerable young patients.

Address correspondence to Mang Shi, shim23@mail.sysu.edu.cn, Yun Guo, guoyun@jiangnan.edu.cn, or Yan-Jun Kang, kangyj@jiangnan.edu.cn.

Yuqi Liao, Jinxia Cheng, and Suwan Xiong contributed equally to this article. The author order was determined by the duration of their research involvement.

The authors declare no conflict of interest.

**KEYWORDS** pediatric pneumonia, infectome, COVID-19, co-infections, metagenomic sequencing, metatranscriptomic sequencing

Community-acquired pneumonia (CAP) is an acute respiratory infection affecting the lungs, including the alveoli and the distal bronchial tree, with various viral, bacterial, and fungal pathogens involved (1). It significantly impacts childhood health, causing approximately 120 million new cases annually and nearly 1 million deaths in children under five (2), making it a major public health concern (3, 4). Pathogens involved in pneumonia in children have been extensively studied using a variety of methodologies, including traditional PCR, qPCR, 16S rRNA metagenomics, total DNA sequencing, total RNA sequencing, and targeted panel sequencing. Previous research identified respiratory syncytial virus (RSV), Rhinovirus, and *Streptococcus pneumoniae* as the primary pathogens in pediatric pneumonia (5, 6), differing significantly from those commonly seen in adults (7, 8). Similar findings were reported by Jain et al. (9), with slight variations in the pathogen profile. RNA-based meta-transcriptomics studies, though relatively limited, provide a richer detection landscape, identifying RNA viruses often missed by DNA-based approaches, as well as DNA viruses, bacteria, and fungi (10). The detection rates of various pathogens exhibited clear seasonal patterns: *S. pneumoniae* and *Haemophilus influenzae* were more prevalent in the spring, *Mycoplasma pneumoniae* in the summer, Rhinovirus, RSV, and Influenza A virus in the autumn, and *Chlamydia pneumoniae*, Boca virus, and Influenza B virus in the winter, showing higher rates compared to other seasons (11). Geographically, southern China has a higher positive rate of viruses compared to northern China, with relatively smaller monthly variations (12). The epidemic dynamics driven by climate factors (mainly temperature) partly explain the overall detection rates and seasonal patterns of respiratory viruses, and these relationships vary with latitude (13).

The COVID-19 pandemic (14, 15), caused by SARS-CoV-2, has significantly disrupted the pathogen profile in pediatric pneumonia. During the pandemic, SARS-CoV-2 became the dominant pathogen, while others declined or even disappeared (16–18), likely due to strict public health measures (19, 20). However, after these restrictions were lifted, the pathogen landscape shifted dramatically, with notable changes in prevalence, intensity, and seasonality (21–26). In China, detection rates of RSV, Influenza A virus, and *M. pneumoniae* surged following the removal of COVID-19 restrictions in December 2022 (27, 28). The age range of susceptible children also widened, and co-infections involving multiple pathogens became increasingly common, often resulting in more severe symptoms and worse outcomes (27, 29, 30). Despite these findings, a comprehensive understanding of the changes in pediatric pneumonia pathogens, co-infection patterns, and secondary infections before and after the pandemic restrictions remains incomplete.

Most of the above-mentioned studies focus primarily on case-based investigations, with limited attention given to disease progression. However, the process of respiratory infection is highly complex. Disease progression studies in respiratory infections have shown that factors such as the host's immune response (31–33), antagonism by commensal microbiota (34–37), co-infections, and opportunistic infections (12, 38, 39) can all significantly influence clinical outcomes. For example, a recent study (40) reveals strong links between upper respiratory tract (URT) microbiome dynamics and disease progression and clinical outcomes. In pediatric cases, such progression-based investigations remain scarce. The limited studies available have revealed that co-infections with viruses and bacteria often lead to more severe disease and longer recovery times (41). However, these studies often lack comprehensive data on the specific role the dynamic changes of the microbiome in shaping disease progression.

In this study, we used both metagenomic and metatranscriptomic sequencing on 116 sputum samples from 58 hospitalized children with CAP in Wuxi, China. These children were enrolled during both pre- and post-COVID-19 restrictions in China. Each child provided two samples to capture the longitudinal dynamics of the respiratory "infectome" throughout the disease course. By comparing the pathogen prevalence and

abundance between the two time periods and sampling points, we aim to unveil the shift of pathogen landscape in pediatric pneumonia related to the COVID-19 pandemic and to the progression of disease in patients. Additionally, our findings will provide valuable insights into the dynamic interactions between the respiratory "Total infectome" and the pathogenesis of pneumonia.

## MATERIALS AND METHODS

### Study design and participants

This prospective cohort study enrolled children hospitalized with CAP at the Department of Respiratory Medicine, Wuxi Children's Hospital. The study covered two time periods: September to December 2022 (during COVID-19 control measures) and April to September 2023 (after relaxation of control measures). The diagnosis of CAP was based on the "Diagnosis and Treatment Guidelines for Community-Acquired Pneumonia in Children (2019 edition)" issued by the National Health Commission of China, which included clinical signs and symptoms (e.g., cough, fever, and tachypnea) and radiographic evidence of pulmonary infiltrates. Children aged 1 month to 10 years without immunodeficiency or chronic diseases were eligible for the study. This study was approved by the Ethics Committee of Wuxi Children's Hospital (NO. WXCH2021-09-004). Informed consent was obtained from the parents or legal guardians of all individual participants included in the study.

### Sample collection and processing

Sputum samples were collected from each patient at two time points: disease phase (first sample) and recovery phase (second sample). The samples were obtained by aspiration using a sterile catheter and immediately placed in DNA/RNA Shield (Zymo Research, USA) for stabilization. Samples were stored at −80℃ until transportation on dry ice to the laboratory at Sun Yat-sen University for further processing. Total DNA and RNA were simultaneously extracted from each sample using the DNA/RNA Miniprep kit (Zymo Research) according to the manufacturer's protocol. The quantity and quality of extracted DNA and RNA were assessed using a Qubit 4.0 Fluorometer (Invitrogen, USA) and an Agilent 2100 Bioanalyzer (Agilent Technologies, USA), respectively.

### Library preparation and sequencing

The Universal Plus DNA Library Prep Kit (Novogene) was used for DNA fragmentation, end repair, adapter ligation, amplification, and purification to construct paired-end libraries. For RNA samples, the Ribo-off rRNA Depletion Kit was used to remove human rRNA sequences. Subsequently, the Universal V8 RNA-seq Library Prep Kit (Novogene, China) was used for cDNA synthesis, fragmentation, end repair, adapter ligation, library amplification, and purification to construct paired-end libraries. The quality of the prepared libraries was assessed using the Qsep 100 bioanalyzer (BiOptic, Taiwan, China). Qualified libraries were then sequenced on the Illumina NovaSeq 6000 platform (Illumina, USA) following the manufacturer's standard protocols, generating 150 bp paired-end reads.

### Bioinformatics analysis

The quality of the raw sequencing reads was assessed using fastp v0.23.2 (42) with default parameters. BBMap v38.62 was then used to remove low complexity reads, followed by the removal of duplicate reads using CD-HIT v4.8.1 (43). To remove rRNA reads, the filtered reads were aligned to the SILVA rRNA database using Bowtie2 v2.4.5 with default settings, resulting in non-rRNA (norRNA) reads (44). Subsequently, the norRNA reads were aligned to the human reference genome GRCh38 using Bowtie2 to obtain norRNA non-human reads for downstream analysis.

The norRNA non-human reads were used for bacterial analysis at both the genus and species levels. Genus-level abundance was quantified by aligning the reads to the rRNA-free GTDB database (https://gtdb.ecogenomic.org) using Bowtie2. For each genus, read counts were retained if the number of aligned reads in a sample exceeded 1,000; otherwise, they were considered false positives. Species-level identification of bacterial pathogens was performed using a phylogenetic approach based on conserved marker gene rpoB. This involved mapping the reads to reference gene sequences, generating consensus sequences, downloading corresponding gene sequences from related species, aligning the sequences using MAFFT v7.505 (45), trimming poorly aligned regions using MEGA v7 (19) and TrimAl v1.4 (20), and constructing maximum-likelihood phylogenetic trees using PhyML (21). Samples clustering with known pathogen species were considered reliable positives. For confirmed bacterial pathogens, the mapped reads were extracted, re-aligned to the rRNA database to remove residual rRNA reads, and quantified as RPM (reads per million total reads) for standardization. Fungal pathogens were identified and quantified using the same phylogenetic approach as described for bacteria, based on conserved marker gene TEF1 and maximum-likelihood tree construction.

Viral analysis involved aligning the norRNA non-human reads to the NCBI viral genome database (nt) using BLAST and the non-redundant protein database (nr) using DIAMOND v2.0.15 (24) in blastx mode. The annotated viruses were further quantified and subjected to phylogenetic analysis for species-level identification. For each identified virus, the norRNA non-human reads were mapped to its reference genome using Bowtie2 to quantify the abundance in RPM, discarding results with RPM <1 (22). Then, we referenced several literatures and "List of Pathogenic Microorganisms Infecting Humans" to retain only those pathogens known to infect humans (12, 46).

## Statistics and reproducibility

Pathogen prevalence was defined as the percentage of patients with RPM >10 in either sample and compared between the 2022 and 2023 cohorts using Fisher's exact test, with 95% confidence intervals calculated by the Wilson score method. The positivity rate for each pathogen was calculated as the percentage of patients with an RPM value >10 in either the first or second sample. The Wilcoxon signed-rank test assessed changes in pathogen abundance between the first and second time points. Categorical variables were presented as numbers and percentages and continuous variables as medians and interquartile ranges (IQRs). Heatmaps and Venn diagrams were used to visualize the dynamics and co-occurrence patterns of pathogens and microbial communities. All analyses were performed using R v4.3.0, with $P$-values < 0.05 considered statistically significant.

## RESULTS

### Characteristics of the study cohort

A total of 63 hospitalized children with pneumonia were initially enrolled in this study, with sputum samples collected twice from each child during their illness. The first sample was obtained during the acute phase, followed by a second sample taken, on average, 3.6 days later (Fig. 1). After excluding four patients with only one sample and one with insufficient DNA/RNA concentration, 58 patients were included in the final analysis. Among these, 31 cases were sampled between September and December 2022, prior to the relaxation of COVID-19 restrictions that led to a large-scale outbreak of SARS-CoV-2 in China. The remaining 27 cases were documented from April to September 2023, after the outbreak. The demographic and clinical characteristics of the patients are summarized in Table 1. The median age was 2.7 years (range: 0.4–9.3 years), and 57.9% of the patients were male. The median duration of hospitalization was 7 days. All patients received antibiotic therapy, and the use of antiviral agents rose from 12.9% in 2022 to 40.7% in 2023. Ultimately, all patients recovered and were discharged from the hospital (Fig. 1).

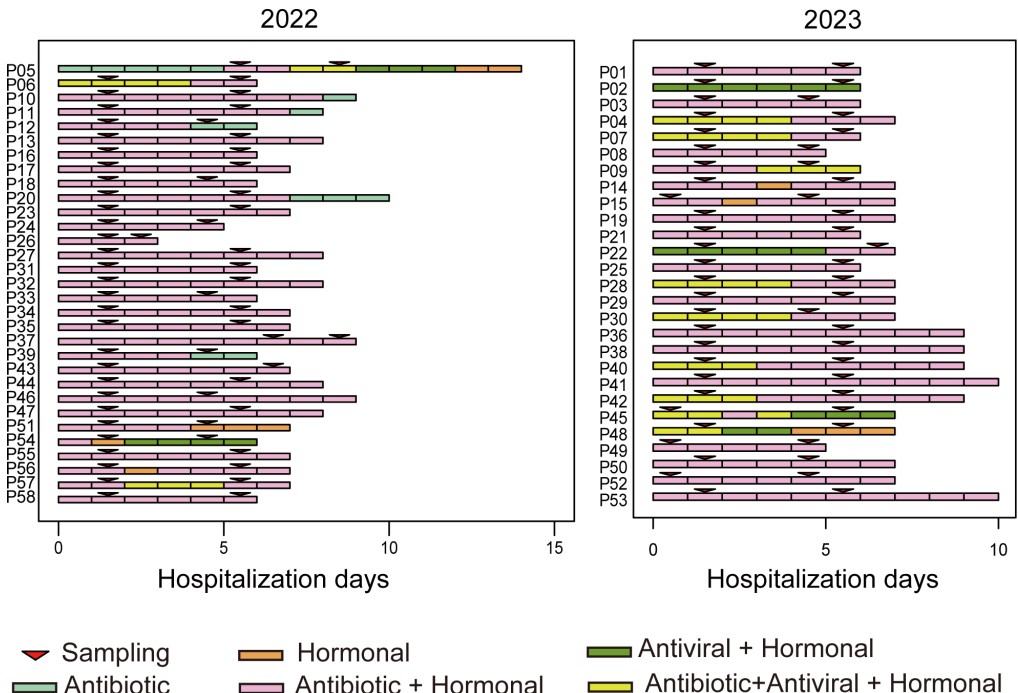

**FIG 1** Summary of CAP disease progression, treatment regimens, and sample collection for 58 patients. All patients are temporally aligned to their day of hospital admission for CAP. The frames indicate days of inpatient care, with colors representing the treatment regimen administered on each day. Sample collection during hospitalization is marked by triangle symbols, with all patients having samples collected at two distinct time points.

## Characterization of sequencing data

We conducted metagenomic and metatranscriptomic sequencing on 116 sputum samples (two per patient), resulting in a total of 232 sequencing libraries. The median number of raw reads per library was 82.4 million (IQR: 70.2–122.8 million). After quality control, host read removal, and rRNA depletion, the median number of high-quality microbial reads was 5.80 million (IQR: 1.71–11.5 million) for metagenomics and 2.88 million (IQR: 1.77–4.33 million) for metatranscriptomics. The proportions of metagenomic and metatranscriptomic reads varied significantly among different libraries (Fig. S1). The raw sequencing data are available in the Sequencing Read Archive database under Bioproject ID PRJNA1176127.

## Diversity and composition of the respiratory infectome

In total, we identified 30 pathogen species across 17 microbial families from all samples (Fig. 2). This included 11 RNA viruses (e.g., SARS-CoV-2, Influenza A virus, and RSV), 6 DNA viruses (e.g., Adenovirus and Cytomegalovirus), 12 bacteria (e.g., *M. pneumoniae*, *H. influenzae*, and *S. pneumoniae*), and 1 fungus (*Candida albicans*). Among the 58 patients in this study, the most prevalent pathogens detected were *M. pneumoniae* (34.48%), *H. influenzae* (32.76%), RSV (20.69%), Rhinovirus A (18.97%), *S. pneumoniae* (12.07%), and Rhinovirus C (8.62%). Additional pathogens with a positivity rate greater than 5% included Influenza A virus (6.90%), Human respirovirus 3 (6.90%), *Haemophilus parainfluenzae* (6.90%), Enterovirus D (5.17%), Human betaherpesvirus 5 (5.17%), Human polyomavirus 4 (5.17%), *Staphylococcus aureus* (5.17%), and *Escherichia coli* (5.17%). Phylogenetic analysis confirmed the species-level identity of the bacterial and fungal pathogens (Fig. 3) and revealed intra-species diversity among the viral pathogens (Fig. 4; Fig. S2). The pathogen sequences characterized in this study are available at NCBI/ GenBank under accession numbers PQ541342 to PQ541397.

TABLE 1 Baseline characteristics of the patients enrolled in this study

| | Cases from 2022 (n = 31) | Cases from 2023 (n = 27) |
|---|---|---|
| Age, year (no. [%]) | | |
| [0–1] | 4 (12.9) | 4 (14.8) |
| (1–3] | 9 (29.0) | 9 (33.3) |
| (3–6] | 16 (51.6) | 9 (33.3) |
| (6–10] | 2 (6.5) | 5 (18.5) |
| Sex (no. [%]) | | |
| Male | 21 (68.7) | 14 (51.9) |
| Female | 10 (32.3) | 13 (48.1) |
| Hospitalization duration, day mean (range) | 7.2 (3–14) | 7.1 (5–10) |
| Sampling time interval, day mean (range) | 3.61 (1–6) | 3.67 (1–5) |
| Treatment (no. [%]) | | |
| Antibiotic treatment | 31 (100) | 26 (96.3) |
| Antiviral treatment | 4 (12.9) | 11 (40.7) |
| Hormonal treatment | 31 (100) | 27 (100) |
| Clinical outcome (no. [%]) | | |
| Discharged | 31 (100) | 27 (100) |

## Changes in the prevalence rates of infectome

We compared the prevalence of respiratory pathogens between patient cohorts from 2022 (during COVID-19 control measures) and 2023 (after the relief of these measures; Fig. 5). The total number of detected pathogens is 70 in 2022 and 73 in 2023. Notable increases were observed for *M. pneumoniae* (5.56%–36.4%, $P = 0.024$), RSV (2.38%–14.5%, $P < 0.001$), and *H. influenzae* (3.97%–14.5%, $P = 0.005$). Conversely, the prevalence of Rhinovirus A decreased from 14.3% to 0% ($P < 0.001$). Changes in the prevalence of other pathogens were not statistically significant.

## Comparisons of metagenomics vs metatranscriptomics approaches

We analyzed the differences in pathogen detection at the species level between metagenomic and metatranscriptomics sequencing approaches, excluding samples where neither approach identified any pathogens. Metatranscriptomics showed clear advantages over metagenomics in detecting RNA viruses (Fig. 6). While it can also identify DNA viruses, it reported significantly lower abundances of DNA viruses (Epstein-Barr virus, Cytomegalovirus, Human Herpesvirus 7 [HHV-7], and Human Polyomavirus 4) compared to metagenomics ($P < 0.05$), suggesting reduced sensitivity for DNA viruses. In terms of bacteria and fungi, *S. pneumoniae* and *C. albicans* exhibited higher abundances in metatranscriptomics sequencing ($P < 0.05$), while other pathogens did not show statistically significant differences (Fig. 6). These results suggest that while metatranscriptomics effectively captures the total infectome, the metagenomic approach enhances the detection sensitivity of DNA viruses. Thus, for the total infectome analysis, we utilized metatranscriptomics results for RNA viruses, bacteria, and fungi, while employing metagenomics for DNA virus data.

## Longitudinal dynamics of the respiratory microbiome during infection

To characterize the changes of the respiratory microbiome during the disease course of pneumonia, we compared microbial abundances between the first and second time points for each patient (Fig. 7). Overall, the abundances of RNA viruses, bacterial pathogens, and fungal pathogens significantly decreased from first to the second time point (all $P < 0.05$; Fig. 7A). In contrast, DNA viruses showed an increasing trend ($P = 0.081$). While the total commensal bacterial abundance remained stable ($P = 0.45$),

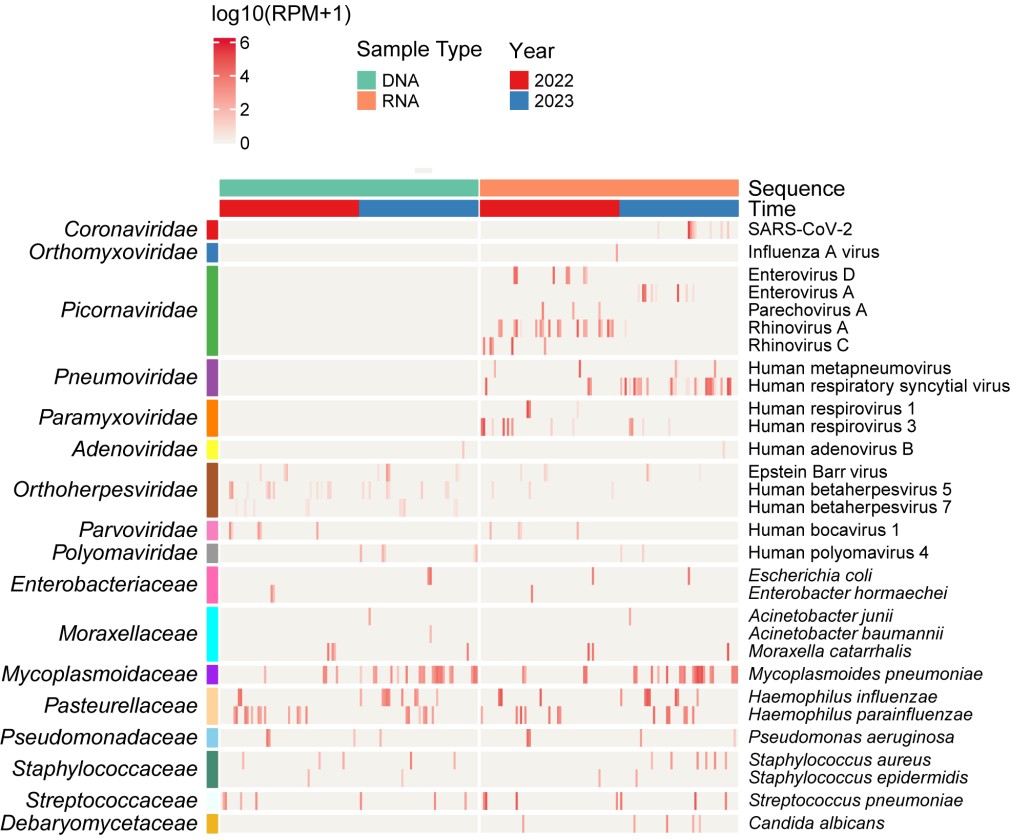

**FIG 2** Heatmap depicting pathogen abundance of viruses and bacteria. The top panel annotations represent grouping variables, including sequencing methods and collection years. The right panel provides species-level taxonomic annotations for identified pathogens, while the left panel summarizes data at the family level. The heatmap blocks are color coded to indicate pathogen abundance.

genera *Clostridium*, *Lactobacillus*, and *Lactococcus* showed significant increased abundance level (all $P < 0.05$; Fig. 7B). Interestingly, when examining individual pathogens, some (e.g., Parechovirus A and *S. pneumoniae*) showed increased abundance or only appeared in the second time point in certain cases (red lines; Fig. 7B). This suggests the possibility of secondary infections or hospital-acquired infections.

## Case-by-case studies

To better depict the dynamics of the respiratory microbiome during the disease course, we analyzed the microbiomes of representative individual cases (Fig. 8).

### *Patient 02: A case of rapid pathogen clearance and clinical recovery*

A 1-year- and 4-month-old girl was admitted on 18 May 2023, presenting with a 4-day history of severe cough and wheezing. Upon admission, she exhibited respiratory distress, evidenced by tachypnea (48 breaths/min) and the inspiratory three-concave sign, with wheezing and coarse rales detected in both lungs. A chest X-ray revealed patchy shadows in both lungs. Initial laboratory tests showed a normal white blood cell count ($7.7 \times 10^9$/L) and a low C-reactive protein (CRP) level (1.5 mg/L), suggesting a non-bacterial etiology. The initial diagnosis was wheezing bronchopneumonia.

The first sputum sample was collected on 19 May. Metatranscriptomic analysis identified human RSV as the primary pathogen, with a low abundance of Rhinovirus A also detected (Fig. 8A). This finding was consistent with the clinical presentation and laboratory results, which showed no evidence of invasive bacterial infection

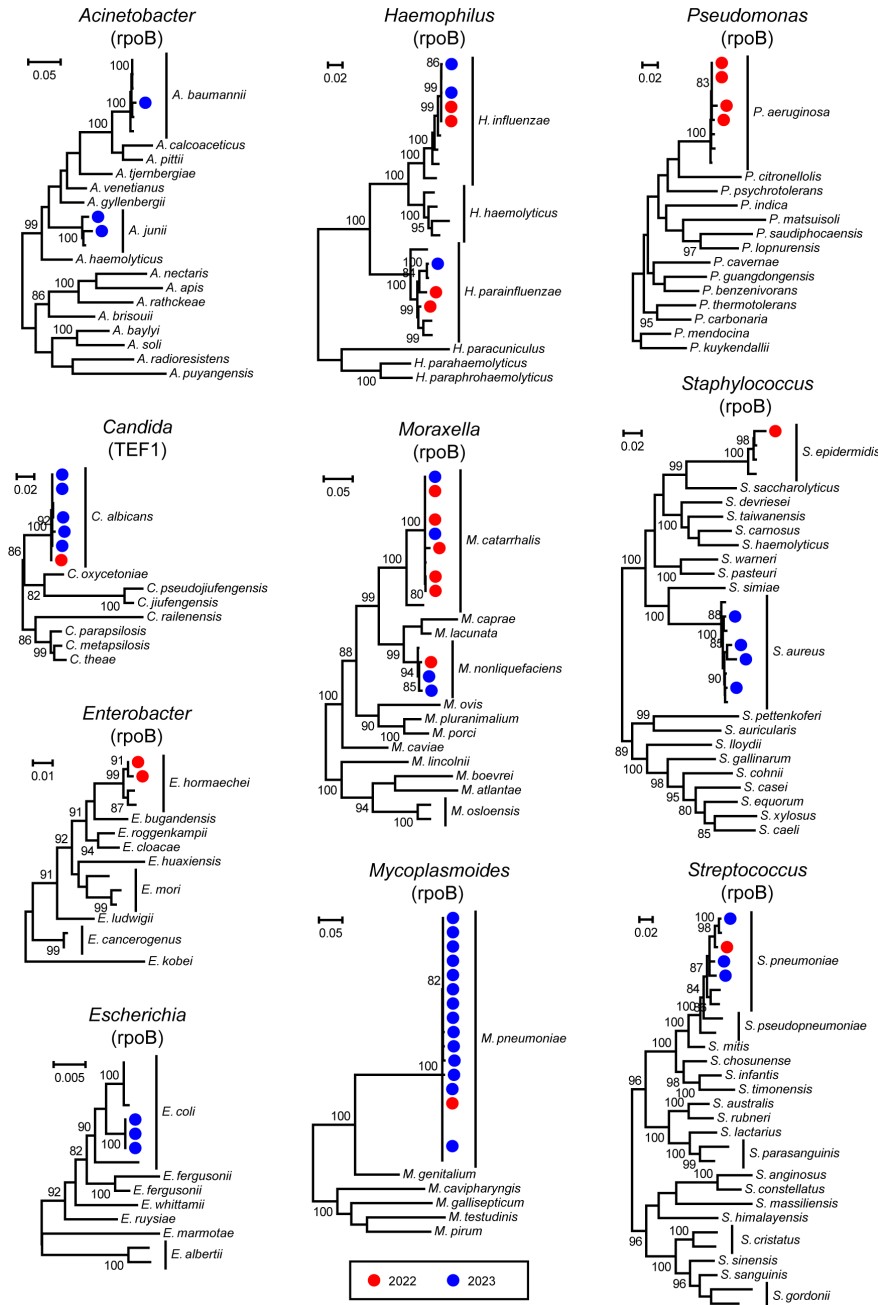

**FIG 3** Phylogenetic trees of detected bacteria and fungi maximum likelihood phylogenetic trees are constructed using conserved genes: RNA polymerase beta subunit (rpoB) for bacteria and translation elongation factor EF-1 alpha (TEF1) for fungi. For clarity, all trees are midpoint-rooted. Red dots denote samples collected in 2022, while blue dots represent samples from 2023.

(procalcitonin [PCT] <0.02 ng/mL). Based on the diagnosis of viral pneumonia, the patient received antiviral therapy (interferon nebulization) and systemic steroids (intravenous methylprednisolone) to control inflammation and alleviate bronchospasm. The patient's clinical condition improved rapidly following treatment. By 20 May, her wheezing and respiratory distress had significantly subsided. A second sputum sample was collected on 23 May, the day of discharge. The follow-up metatranscriptomic analysis showed that the abundances of both RSV and Rhinovirus A had dropped to undetectable levels (Fig. 8A). This microbiological clearance directly correlated with her full clinical recovery. This case exemplifies an ideal clinical course where accurate

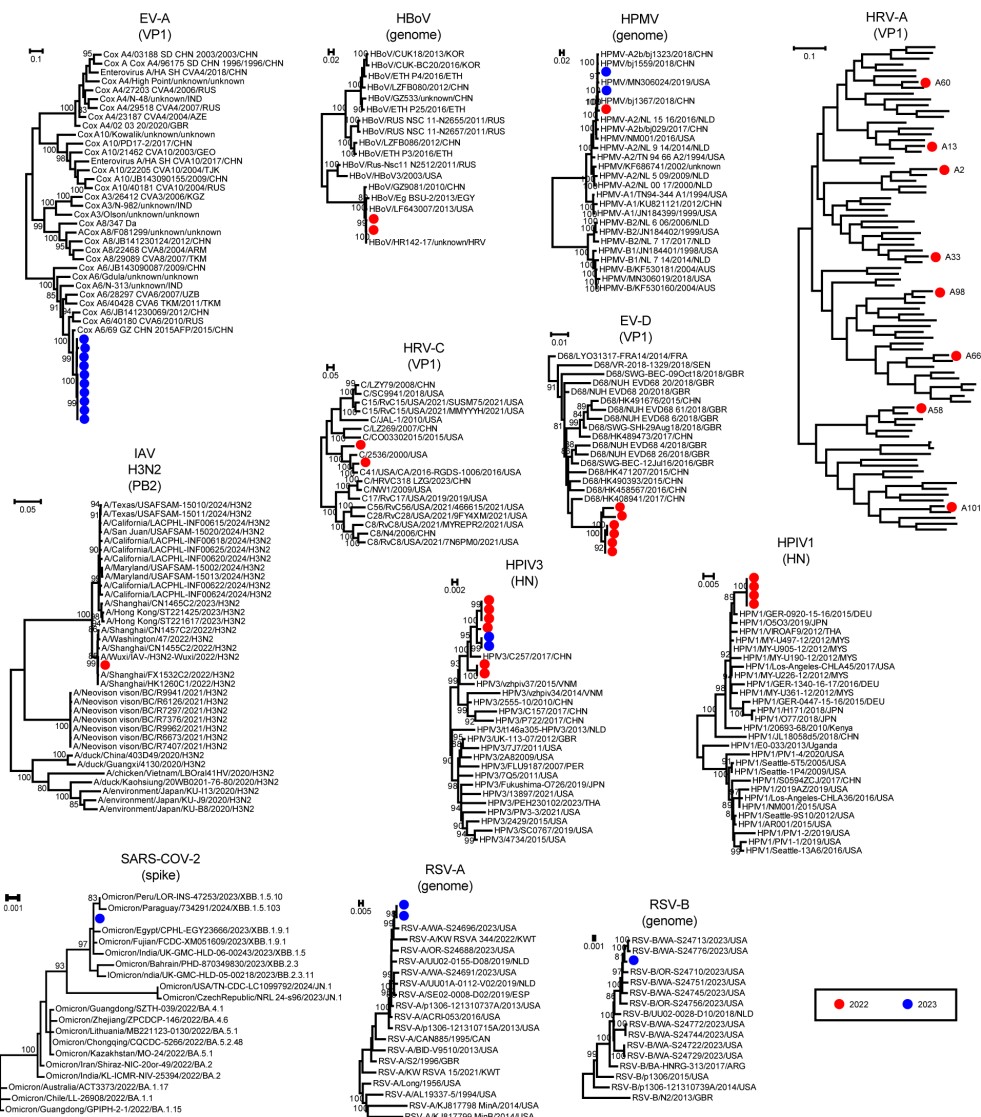

**FIG 4** Phylogenetic trees of detected viruses. Maximum likelihood phylogenetic trees are constructed using viral genomes or marker genes, such as capsid protein 1 (VP1), envelope glycoprotein gB (gB), polymerase PB2 (PB2), surface glycoprotein (spike), and hemagglutinin-neuraminidase protein (HN). All trees are midpoint-rooted for clarity. Red dots represent samples collected in 2022, while blue dots indicate samples from 2023.

pathogen identification via metatranscriptomics guided effective antiviral and anti-inflammatory treatment, leading to swift pathogen eradication and resolution of symptoms.

### Patient 04: A case of complex co-infection

A 7-month-old boy was admitted on 29 August 2023, with an 8-day history of cough that had worsened with wheezing for the past 3 days. Prior to admission, he had received courses of erythromycin and cefuroxime without significant improvement. On admission, he was afebrile but tachypneic (40 breaths/min), with wheezing and coarse rales audible in both lungs. A chest X-ray showed patchy infiltrates in the left lower lobe. Notably, his CRP level had decreased from 13.2 mg/L a week prior to 0.6 mg/L upon admission, while his platelet count was elevated ($586 \times 10^9$/L), suggesting a complex inflammatory response. Routine serology for common respiratory pathogens, including M. pneumoniae and RSV, was negative.

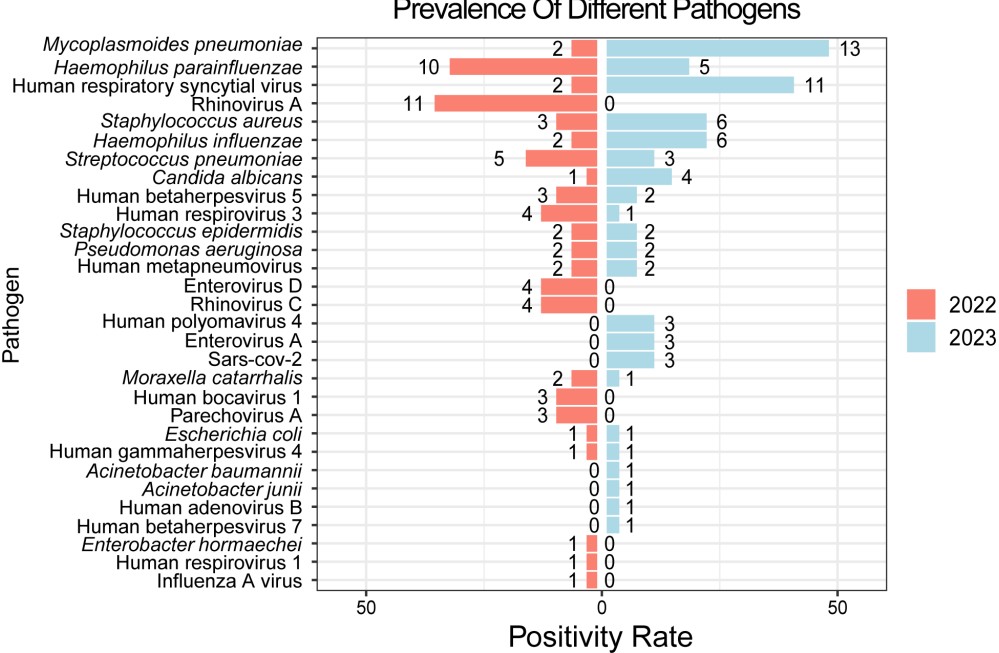

**FIG 5** Prevalence of different pathogens before and after the release of COVID restrictions. The colors represent different sampling time groups. Bar lengths correspond to pathogen positivity rates, and the numbers indicate the number of infected patients, arranged in descending order based on overall prevalence.

The initial sputum sample, collected upon admission, revealed a complex co-infection profile through metatranscriptomic sequencing that was not captured by conventional tests. It identified *M. pneumoniae* and human RSV as the primary pathogens, alongside a significant presence of *C. albicans* (Fig. 8B). This finding explained the patient's persistent symptoms despite prior antibiotic therapy. Consequently, a comprehensive treatment regimen was initiated, including broad-spectrum antibiotics (amoxicillin-clavulanate), antiviral therapy (interferon), and systemic steroids. A dermatology consult confirmed a diagnosis of cutaneous candidiasis, consistent with the high abundance of *C. albicans* detected in the sputum.

By 31 August, the patient's wheezing and respiratory symptoms had partially improved. A second sputum sample collected on this day showed a dynamic shift in the infectome. While RSV abundance had decreased and *C. albicans* was cleared, the abundance of M. pneumoniae remained high. Critically, the microbial landscape shifted, marked by the emergence of transcriptionally active *Staphylococcus epidermidis* (Fig. 8B), a known opportunistic pathogen. This finding, suggesting a potential secondary bacterial complication, supported the decision to continue antibiotic treatment to prevent clinical deterioration. The patient was eventually discharged with medication on 4 September. This case highlights the limitations of conventional diagnostics in identifying complex co-infections. It demonstrates how longitudinal metatranscriptomic monitoring can uncover the dynamic interplay between primary pathogens, opportunistic fungi, and subsequent secondary bacterial infections, providing critical information for adjusting treatment strategies and managing prolonged pneumonia.

### Patient 17: A case of viral-driven exacerbation with secondary bacterial infections

A 5-year- and 8-month-old boy with a history of allergic rhinitis and recurrent asthma was admitted on 24 September 2022, for an acute exacerbation, presenting with a 3-day history of cough and wheezing that had progressed to dyspnea. On admission, he was

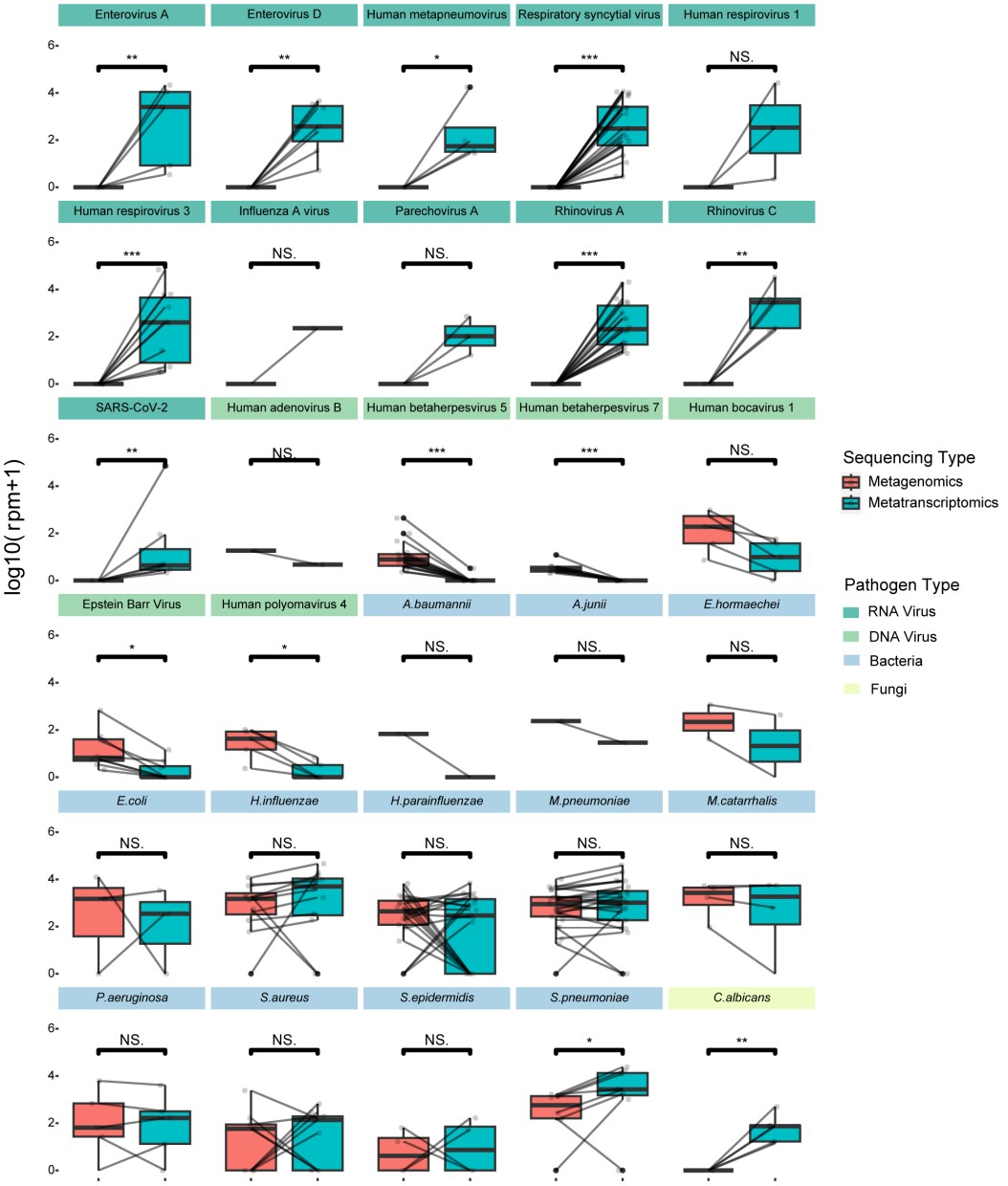

**FIG 6** Abundance differences of detected pathogens using DNA or RNA sequencing methods facet title background colors correspond to different pathogen types. Lines connect results for the same sample from the two sequencing methods. Differences between groups are analyzed using the Wilcoxon rank-sum test. Statistical significance is denoted as *: $P \leq 0.05$, **: $P \leq 0.01$, ***: $P \leq 0.001$, and NS indicates not significant.

tachypneic (36 breaths/min) with widespread wheezing audible in both lungs. Initial blood tests revealed a notable neutrophilia (WBC $10.76 \times 10^9$/L, 92.8% neutrophils) and an elevated heparin-binding protein (HBP) level (62.98 ng/mL), suggesting a significant inflammatory, possibly bacterial, process. A chest X-ray showed bronchial wall thickening and a blurry shadow in the left lung. Based on these findings, he was diagnosed with a severe asthma exacerbation and bronchopneumonia and was started on antibiotics (cefuroxime) and systemic steroids.

The first sputum sample, collected upon admission, provided a different perspective through metatranscriptomic analysis. It revealed a predominant multi-viral infection, with high abundances of Enterovirus D and Rhinovirus A, alongside HHV-7 (Fig. 8C). In

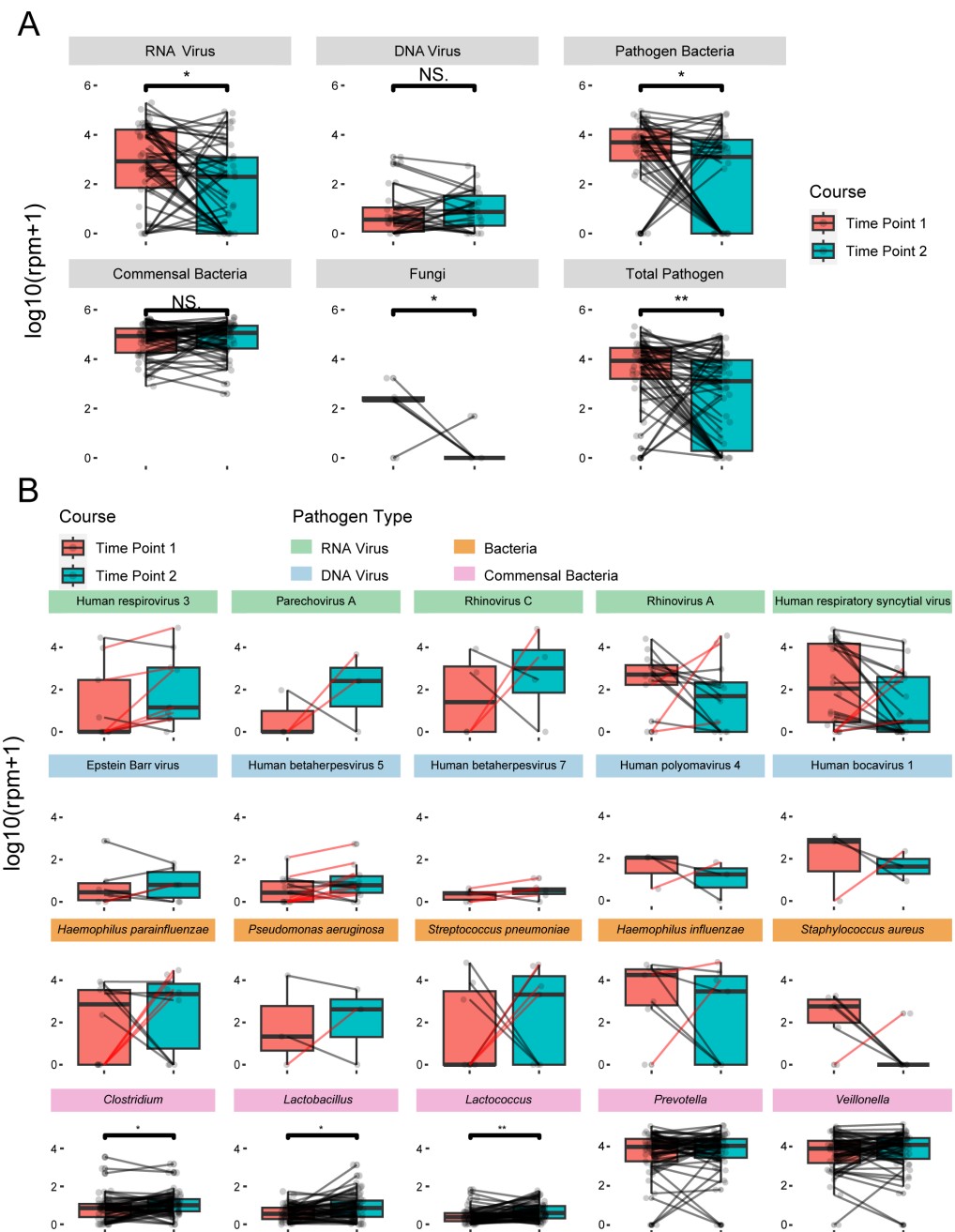

**FIG 7** Dynamics of respiratory infectome during the course of CAP hospitalization. (A) Microbiome abundance at different clinical stages of disease. Differences between groups were analyzed using the Wilcoxon rank-sum test. (B) Microorganisms showing increasing abundance trends at different clinical stages. Lines connect two sampling results from the same patient, with red indicating an increase in pathogen abundance.

contrast, the only bacterium detected by the hospital's sputum culture was *Klebsiella pneumoniae*. This discrepancy suggests that the patient's severe symptoms were primarily driven by a viral infection, which may have been inadequately addressed by the initial antibiotic-focused regimen. The patient's condition improved with steroid and bronchodilator treatments, and he was discharged on 30 September. A second sample was collected on 29 September, prior to discharge. At this time point, while HHV-7 was cleared and the patient's wheezing had resolved, both Enterovirus D and Rhinovirus A persisted at high levels. Furthermore, the analysis revealed the emergence of secondary bacterial pathogens, specifically *S. pneumoniae* and *H. parainfluenzae* (Fig. 8C).

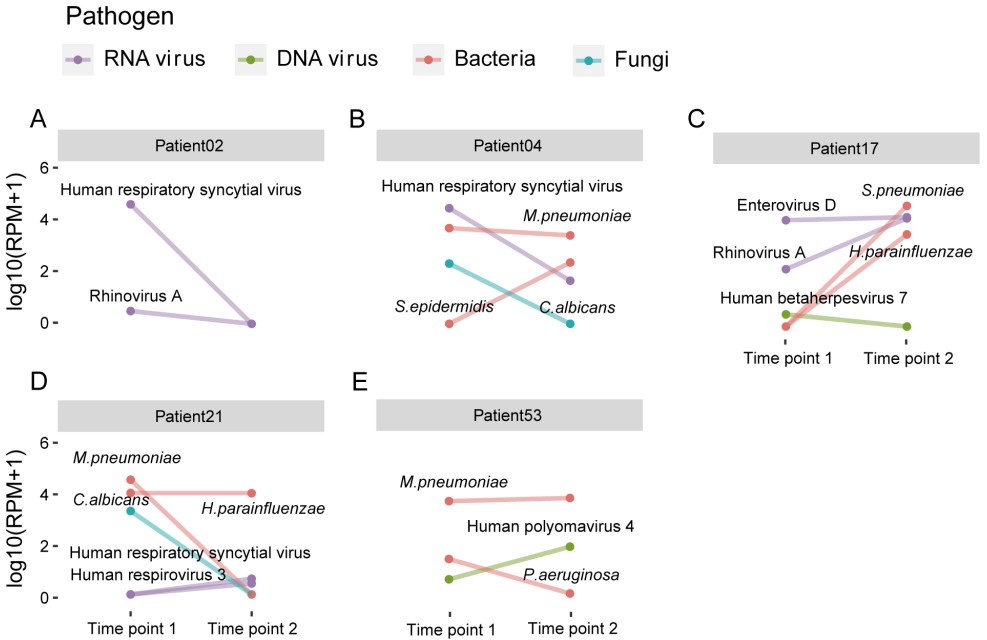

**FIG 8** Longitudinal dynamics of the respiratory infectome in five representative pediatric pneumonia cases. The abundance of pathogens at two distinct time points (Time point 1: upon admission; Time point 2: during follow-up or at discharge) is shown. Different colors represent pathogen types. Each panel highlights a distinct clinical and microbiological scenario: (A) Patient 02, illustrating rapid viral clearance (RSV) correlating with full clinical recovery. (B) Patient 04, showing pathogen persistence (M. pneumoniae) and secondary infection (S. epidermidis) during a prolonged clinical course. (C) Patient 17, demonstrating persistent viral drivers and the emergence of secondary bacterial pathogens, a profile missed by conventional diagnostics. (D) Patient 21, illustrating pathogen succession, where clearance of the primary pathogen was followed by the dominance of a pre-existing bacterium and emergence of new viruses. (E) Patient 53, showing how an unresolved microbial burden (persistent M. pneumoniae and an emergent virus) correlated with clinical deterioration requiring bronchoscopy.

This case illustrates a scenario where the primary drivers of illness—multiple respiratory viruses—were missed by conventional diagnostics.

### Patient 21: A case illustrating the sequential emergence of pathogens

A 4-year- and 10-month-old girl was admitted on 31 August 2023, with a 5-day history of high fever (up to 40°C) and cough. A chest X-ray taken prior to admission revealed right upper lobe pneumonia. She had already received a 3-day course of intravenous amoxicillin-clavulanate, which had partially reduced her fever. Upon admission, her inflammatory markers were significantly elevated (CRP 20.1 mg/L, HBP 98.82 ng/mL, and IL-6 86.67 pg/mL), indicating a strong ongoing inflammatory response. The initial diagnosis was bronchopneumonia.

The first sputum sample, collected upon admission, was analyzed using metatranscriptomics. It revealed a polymicrobial infection dominated by *M. pneumoniae*, with co-infections of *C. albicans* and *H. parainfluenzae* also detected (Fig. 8D). This finding prompted a switch in antibiotic therapy to azithromycin to target the atypical pathogen, supplemented with oral steroids to control inflammation. The patient's clinical condition improved, and her fever subsided by 3 September. A second sputum sample was collected on 4 September, the day of discharge. The follow-up analysis revealed a dramatic shift in the infectome. While *M. pneumoniae* and *C. albicans* were successfully cleared, *H. parainfluenzae*, which was a minor component initially, had become the dominant pathogen. Furthermore, two new respiratory viruses, human RSV and human respirovirus 3, emerged, although the patient was clinically improving at this stage (Fig. 8D). This case demonstrates the dynamic and sequential nature of respiratory infections. The emergence of new pathogens during clinical recovery suggests that while the

primary pathogenic driver (*M. pneumoniae*) was controlled, the respiratory microbiome remained in a volatile state of succession. The newly detected organisms likely represented transient colonizers or subclinical infections that were being effectively managed by the patient's recovering immune system.

### Patient 53: A case of severe co-infection leading to a complicated clinical course

A 2-year- and 2-month-old boy was admitted on 3 August 2023, with a 5-day history of refractory high fever (up to 40°C) and cough. He had failed to respond to initial antibiotic therapy (cefuroxime). Upon admission, laboratory tests painted a picture of a severe bacterial infection, with a markedly elevated PCT of 1.01 ng/mL, an HBP level of 207.89 ng/mL, and a CRP of 16.2 mg/L. A chest X-ray showed a patchy shadow in the left lung.

Metatranscriptomic analysis of the initial sputum sample, collected upon admission, provided a definitive etiological diagnosis, revealing a severe co-infection with *M. pneumoniae* and *Pseudomonas aeruginosa* (Fig. 8E). This finding was pivotal, as it explained both the severe inflammatory state and the treatment failure. Crucially, *M. pneumoniae* lacks a cell wall and is intrinsically resistant to the beta-lactam antibiotics the patient had received (both cefuroxime and the subsequent amoxicillin-clavulanate). While the escalated antibiotic regimen could target *P. aeruginosa*, the persistence of the resistant *M. pneumoniae* explains the patient's complicated course. The treatment was therefore escalated to a broader-spectrum antibiotic (amoxicillin-clavulanate) and systemic steroids to control the hyper-inflammation. Despite these interventions, the patient's recovery was complicated. A second sputum sample, collected on 8 August, showed that while *P. aeruginosa* had been successfully cleared, the abundance of *M. pneumoniae* remained high. Critically, an infection with Human polyomavirus 4 emerged at this time point (Fig. 8E). This unresolved microbial burden likely contributed to the patient's worsening inflammation, which necessitated a bronchoscopy and alveolar lavage on 9 August for further treatment. Following this intensified intervention, his condition finally began to improve by 12 August. This case powerfully demonstrates how an initial severe bacterial co-infection can prolong hospitalization and complicate recovery.

### Characterization of co-infections and pathogen interactions

Viral-bacterial co-infections were commonly observed among the patients (Fig. 9). The most prevalent type was bacteria-RNA virus co-infection, accounting for 31.1% (33 out of 106 cases), followed by RNA virus-DNA virus-bacteria co-infections at 13.2% (14 out of 106; Fig. 9A). Focusing on *M. pneumoniae* infection, we found that it was most frequently co-detected with RNA viruses (45.8%, 11 out of 24 *M. pneumoniae* infection cases), particularly RSV (33.3%, 8 out of 24 cases; Fig. 9B). The co-occurrence pattern suggests potential synergistic interactions between *M. pneumoniae* and respiratory viruses in the pathogenesis of pneumonia.

### DISCUSSION

In this study, we combined metagenomic and metatranscriptomic sequencing to characterize the respiratory infectome, a dual-omics strategy that leverages the distinct strengths of each method. Metatranscriptomics, by targeting RNA transcripts, offers a view into the functional activity of the microbiome. It is therefore exceptionally sensitive for detecting actively replicating RNA viruses, whose genomes serve as templates for high-level transcription, and for assessing the metabolic state of bacteria (47). In contrast, metagenomics quantifies genomic DNA, reflecting the presence of an organism regardless of its transcriptional activity. This makes it inherently more sensitive for detecting DNA viruses, particularly those in a latent or low-transcription state, which may be underestimated or missed entirely by metatranscriptomics. Our results, which show lower sensitivity of metatranscriptomics for several DNA viruses, are consistent

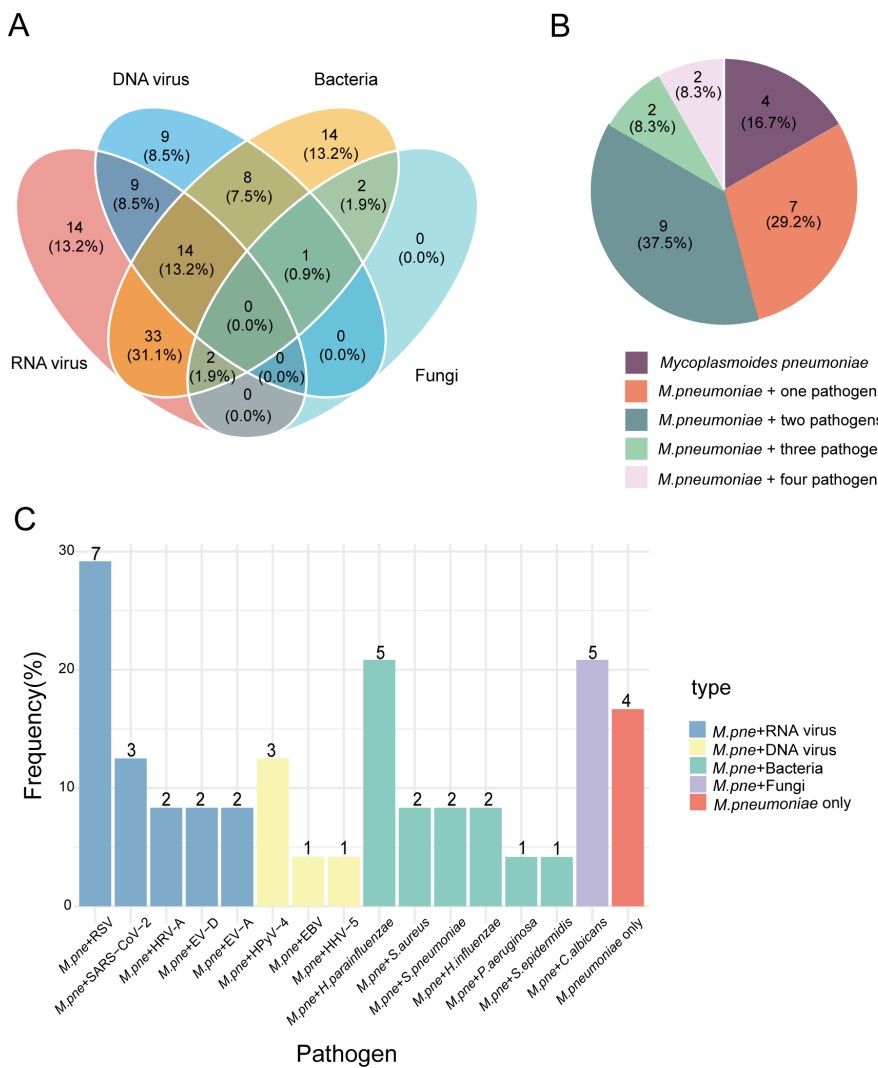

**FIG 9** Co-infections during the course of CAP. (A) Venn diagram of co-infection with different pathogen types. (B) The number of pathogens co-infecting with *M. pneumoniae*. (C) Co-infection of *M. pneumoniae* with other species-level pathogens.

with these underlying principles and reaffirm the value of combining both methods for comprehensive pathogen surveillance (20–22, 24, 25).

Furthermore, our longitudinal analysis revealed a noteworthy dynamic concerning DNA viruses. While the abundances of most RNA viruses and bacterial pathogens significantly decreased over the course of hospitalization—a change likely attributable to effective antiviral and antibiotic therapies—DNA viruses exhibited an opposing, albeit not statistically significant, increasing trend ($P = 0.081$). We hypothesize that this trend may reflect the reactivation of latent DNA viruses, such as members of the *Herpesviridae* and *Polyomaviridae* families detected in our cohort (48, 49). The primary acute infection can induce a state of transient immune dysregulation, which may be further modulated by the administration of corticosteroids—a common treatment in our patient group. This altered immune environment could permit latent viruses to switch from a dormant to a lytic replication cycle, thereby increasing their DNA load at the later time point (50). This observation suggests that the dynamics of the DNA virome could serve as a potential indicator of the host's immune status during recovery from pneumonia (51).

In this study, we observed notable differences in pathogen prevalence between the 2022 and 2023 CAP cohorts, likely influenced by the impact of COVID-19 restrictions. In 2022, less virulent pathogens and opportunistic organisms like Rhinovirus and *H.*

*parainfluenzae* showed higher detection rates, likely due to the suppressive impact of non-pharmaceutical interventions on the transmission of more virulent pathogens, such as Influenza A virus and RSV (27, 29, 30, 34, 35). Following the easing of COVID-19 restrictions, our study documented a sharp resurgence of virulent pathogens, most notably *M. pneumoniae* and RSV, which re-emerged with off-season peaks and heightened intensity. This trend is particularly striking when contextualized by their pre-pandemic epidemiology. Prior to 2020, extensive research had established both *M. pneumoniae* and RSV as leading etiological agents of pediatric CAP, each characterized by predictable seasonal circulation patterns (52, 53). The intense resurgence observed in our 2023 cohort strongly aligns with the "immunity debt" hypothesis, a phenomenon reported globally where populations experience increased susceptibility to endemic pathogens following a prolonged period of limited exposure (6, 36). Our findings, therefore, suggest that the non-pharmaceutical interventions not only suppressed transmission temporarily but also fundamentally altered the established epidemiological dynamics of these key respiratory pathogens, leading to the widespread outbreaks observed upon the relaxation of control measures (37, 38, 40, 54, 55). Consequently, the simultaneous circulation of a diverse array of respiratory pathogens, coupled with frequent co-infections (23, 26–28) such as the co-infection of RSV and *M. pneumoniae*, likely contributed to the severe and prolonged respiratory disease outbreak among children in China during the autumn-winter season.

By comparing the respiratory infectome at hospital admission and subsequent time points, we identified two distinct patterns. In the first pattern, pathogens detected at the initial time point decreased in abundance or disappeared over time, accompanied by an increase in normal flora. Patients exhibiting this trend typically showed good recovery. In contrast, the second pattern, observed in the majority of cases (30/58 cases), involved the detection of new pathogens or opportunistic pathogens at the later time point, often with mixed infections. While some of these patients recovered, many experienced prolonged pneumonia or required additional treatments. Secondary pathogens likely came from either the patient's own microbiota (e.g., *H. parainfluenzae* and Human polyomavirus 4) or the environment, possibly from other hospitalized patients (e.g., *S. pneumoniae* and RSV). Because secondary or co-infections often worsen symptoms (40), future research should focus on better understanding these infection dynamics to improve patient outcomes.

Microbial interactions play a pivotal role in shaping the respiratory microbiome and influencing disease outcomes (56–58). A pivotal finding of our study was the high prevalence of viral-bacterial co-infections, with *M. pneumoniae* and RSV being the most common combination, and this observation aligns with previous studies highlighting the clinical significance of this combination in CAP and underscores the need for a holistic perspective (59, 60). While the precise mechanisms underlying this synergy are not yet fully elucidated, several potential pathways have been proposed based on clinical observations and mechanistic studies of viral-bacterial co-infections. The synergy is likely multifactorial. The key mechanisms appear to involve: (i) virally induced damage to the respiratory epithelium, which enhances bacterial adherence; (ii) potential immunosuppression by *M. pneumoniae*, which facilitates viral infection; and (iii) a critically dysregulated and amplified host inflammatory response (61–63). While clinical evidence strongly supports the existence of this synergy, further mechanistic studies are essential to fully delineate the molecular interactions and to inform the development of more effective therapeutic strategies for managing severe co-infections.

Our longitudinal study design, involving two sampling points, offered a unique window into the dynamic nature of the respiratory infectome during hospitalization. We observed that the abundances of most RNA viruses and bacterial pathogens significantly decreased from the first to the second time point (Fig. 7A). This trend is plausibly a direct reflection of the efficacy of the antibiotic and antiviral therapies administered to the patients, underscoring the critical importance of the initial sample for accurately identifying the primary etiological agents. Concurrently, the follow-up sample provides

invaluable information beyond simple pathogen detection; it serves as a crucial indicator for monitoring treatment response and offers key insights into the emergence of new potential pathogens or shifts in microbial composition that may complicate the clinical course.

Interestingly, a similar decline was noted for fungal abundance, particularly for *Candida*. Given that specific antifungal therapy was not systematically administered, this reduction likely results from a different mechanism. The decline in *Candida* is more likely attributable to the restoration of host immune competence as the patients' overall clinical condition improved. Furthermore, it is plausible that in many of these cases, *Candida* represents a transient colonizer rather than an active pathogen, whose abundance would naturally wane as the primary pathogenic drivers are cleared and the local inflammatory microenvironment is normalized (64, 65). Collectively, these dynamic changes highlight the value of the follow-up sample not just for pathogen detection, but as a crucial indicator for monitoring therapeutic response and understanding the complex ecological shifts within the recovering lung.

Another notable dynamic observed during the course of hospitalization was the significant increase in the relative abundance of certain commensal bacterial genera, namely *Clostridium*, *Lactobacillus*, and *Lactococcus*, while the total commensal abundance remained stable. This phenomenon can be interpreted through the lens of ecological theory, specifically as a case of microbial succession and niche repopulation. During the acute phase of pneumonia, the respiratory ecosystem is dominated by the high burden of primary pathogens, leading to a state of dysbiosis (66). The administration of antibiotics, while effective in reducing the pathogenic load, creates a profound perturbation that disrupts this established community structure, effectively creating a microbial vacuum or vacant ecological niches (67). In the subsequent recovery phase, the respiratory microbiome begins a process of remodeling. Genera such as *Clostridium* and *Lactococcus*, which may be intrinsically more resilient to the administered antibiotics or are opportunistic colonizers originating from the oropharynx or even the gut microbiome, can then proliferate to occupy this newly available ecological space. The rise of these specific genera is therefore not a random occurrence but rather a directed ecological shift (67, 68). This finding is a testament to the resilience and plasticity of the respiratory microbiome and highlights the complex community-level changes that accompany the transition from active disease to clinical recovery. Further exploration of these mechanisms will require in-depth *in vitro* and *in vivo* studies, complemented by multi-omics approaches (e.g., transcriptomics, metabolomics, and immunomics) to integrate data from both the host and microbial communities.

Our study has several limitations. First, the sample size is relatively small, and all samples were collected from a single hospital in Wuxi, which may limit the generalizability of our findings to other regions or patient populations. Second, we only obtained two samples per patient, which may not provide a complete picture of the temporal dynamics of the respiratory microbiome throughout the disease course. Third, the absence of healthy controls prevents direct comparisons between diseased and healthy states. Future studies involving larger, more diverse cohorts, more frequent sampling, inclusion of healthy controls, and comprehensive clinical data are essential to validate and expand upon our findings.

## ACKNOWLEDGMENTS

We would like to thank Gengyan Luo for helpful advice regarding the phylogenetic analysis.

M.S., Y.G., and Y.-J.K.: designed and supervised the study. Y.-Q.L. and J.-X.C: performed the experiments and data analyses. S.-W.X. and Y.-J.K.: collected the specimens and data. S.-W.X., Y.-S.L., J.Q, Y.G., and Y.-J.K.: provided assistance and advice on the project. Y.-Q.L., M.S., and Y.-J.K.: drafted the manuscript. Y.-Q.L. and J.-X.C.: revised the manuscript and provided critical comments. All authors reviewed the results, commented on the manuscript, and approved the final version.

This work was supported by Shenzhen Science and Technology Program (KQTD20200820145822023), Hong Kong Innovation and Technology Fund (ITF) (MRP/071/20X), Guangdong Province "Pearl River Talent Plan" Innovation, Entrepreneurship Team Project (2019ZT08Y464), and the Fund of Shenzhen Key Laboratory (ZDSYS20220606100803007), Top Talent Support Program for young and middle-aged people of Wuxi Health (HB2023090), the Wuxi Taihu Lake Talent Plan (DJTD202304), the Natural Science Foundation of Jiangsu Province (BK20230189), and the Top Talent Support Program for young and middle-aged people of Wuxi Health Committee (BJ2023089).

## AUTHOR AFFILIATIONS

[1]State key laboratory for biocontrol, Shenzhen Key Laboratory of Systems Medicine for inflammatory diseases, School of Medicine, Shenzhen campus of Sun Yat-sen University, Sun Yat-sen University, Shenzhen, China

[2]Guangdong Provincial Center for Disease Control and Prevention, Guangzhou, China

[3]Department of Respiratory Medicine, Wuxi Children's Hospital, The Affiliated Wuxi People's Hospital of Nanjing Medical University, Wuxi, China

[4]Department of Respiratory Medicine and Clinical Allergy Center, Wuxi Children's Hospital, Affiliated Children's Hospital of Jiangnan University, Wuxi, China

[5]Pediatric Laboratory, Wuxi Children's Hospital, Affiliated Children's Hospital of Jiangnan University, Wuxi, China

## AUTHOR ORCIDs

Mang Shi  http://orcid.org/0000-0002-6154-4437
Yun Guo  http://orcid.org/0000-0002-8107-0507
Yan-Jun Kang  http://orcid.org/0000-0002-3169-7037

## AUTHOR CONTRIBUTIONS

Yuqi Liao, Data curation, Formal analysis, Investigation, Software, Validation, Visualization, Writing – original draft, Writing – review and editing | Jinxia Cheng, Formal analysis, Software, Validation, Visualization, Writing – review and editing | Suwan Xiong, Data curation, Resources | Yanshan Liu, Data curation | Jun Qian, Data curation | Mang Shi, Conceptualization, Funding acquisition, Methodology, Project administration, Resources, Supervision, Writing – original draft | Yun Guo, Conceptualization, Methodology, Project administration, Supervision | Yan-Jun Kang, Conceptualization, Funding acquisition, Methodology, Project administration, Resources, Supervision, Writing – original draft

## ADDITIONAL FILES

The following material is available online.

### Supplemental Material

**Figures S1 and S2 (Spectrum01450-25-s0001.pdf).** Fig. S1: Percentage of Reads in Each Library Percentage of reads in both metagenomic and meta-transcriptomic sequencing data of each library. Colors correspond to different types of reads. Fig. S2 Phylogenetic trees of detected viruses. Maximum likelihood phylogenetic trees are constructed using viral genomes or marker genes, such as capsid protein 1 (VP1), envelope glycoprotein gB (gB), and fiber protein (Fiber). All trees are midpoint-rooted for clarity. Red dots represent samples collected in 2022, while blue dots indicate samples from 2023.

Open Peer Review

**PEER REVIEW HISTORY (review-history.pdf).** An accounting of the reviewer comments and feedback.

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
