## [Reviewer comments · Microbiology Spectrum]

Microbiology Spectrum

Dynamics of the Respiratory Infection in Children with Community-Acquired Pneumonia: Insights from Large and Short Time-Scale Analyses

Yu-Qi Liao, Jinxia Cheng, Su-Wan Xiong, Yanshan Liu, Jun Qian, Mang Shi, Yun Guo, and Yan-Jun Kang

Corresponding Author(s): Yan-Jun Kang, Pediatric Laboratory, Affiliated Children's Hospital of Jiangnan University, Wuxi Children's Hospital, Wuxi, China

Review Timeline:

Submission Date:	May 11, 2025
Editorial Decision:	June 17, 2025
Revision Received:	July 16, 2025
Editorial Decision:	August 21, 2025
Revision Received:	September 1, 2025
Editorial Decision:	September 12, 2025
Revision Received:	September 14, 2025
Accepted:	September 25, 2025

Editor: Ana Cabrera

Reviewer(s): The reviewers have opted to remain anonymous.

Transaction Report:

DOI: <https://doi.org/10.1128/spectrum.01450-25>

Re: Spectrum01450-25 (**Dynamics of the Respiratory Infectome in Children with Community-Acquired Pneumonia: Insights from Large and Short Time-Scale Analyses**)

Dear Dr. Yan-Jun Kang:

Thank you for the privilege of reviewing your work. Below you will find my comments, instructions from the Spectrum editorial office, and the reviewer comments.

While we are willing to consider a revised version of this paper at Spectrum, it would be in your best interest to improve the writing. I recommend that you ask a colleague of yours who is a native English speaker to read and provide you some feedback on the writing. You are also welcome to use one of the services here: <https://journals.asm.org/writing-your-paper#language-editing-services>

Revision Guidelines

Sincerely,
Ana Cabrera
Editor
Microbiology Spectrum

Reviewer #1 (Comments for the Author):

Overall Evaluation

This cohort study integrated dual-omics approaches (metagenomics and metatranscriptomics) and infectome profiling of 116 sputum samples from 58 pediatric patients, elucidates the dynamic shifts in the pathogen profile of childhood community-acquired pneumonia (CAP). It systematically records the temporal changes in pathogen abundance and detection rate before and after the relaxation of COVID-19 prevention and control measures, highlighting the prevalent virus-bacterial co-infection patterns. These findings provide critical insights for precise clinical management and evidence-based public health policies for pediatric CAP.

Major Concerns:

Just as the author mentioned in the Discussion, only samples from two time points were collected, and the sample size was not large enough to explain the dynamic changes of the pathogen comprehensively. Please discuss more whether the author's selection of the sampling time excluded the interference of other factors, in other words, statistically robust.

Minor concerns:

In the discussion section of the manuscript, the author has written many established facts, such as lines 390-393, which seem somewhat redundant. It is suggested to simplify some sentences to make the conclusion appear more explicit.

Specific Comments:

1. Line 160-161, please add a citation for the fastp software.

2. Line 233-234, the authors identified "30 pathogen species across 17 microbial families" from all samples. Does this represent all microbes in these samples or a filtered result? If non-pathogenic microorganisms were excluded, what criteria and methods were used to screen for pathogens?

3. Line 233-236, We note discrepancies between DNA and RNA sequencing results in Figure 2. For example, several samples detected Human betaherpesvirus 7 via DNA sequencing but not RNA sequencing, whereas *Candida albicans* showed the opposite pattern. Please explain these differences.

4. Line 245, Figure 4 contains numerous sub-figures that compromise readability. We recommend retaining only representative phylogenetic trees in the main figure and moving the remainder to supplementary materials.

5. line 288-289, What does the vertical axis represent? Is it $\log_{10}(\text{RPM}+1)$? Please supplement this information.

Reviewer #2 (Comments for the Author):

This is a prospective cohort study, which analyzes the NGS results of sputum from children before and after COVID-19 restrictions, providing certain clinical significance.

This is a prospective cohort study, which analyzes the NGS results of sputum from children before and after COVID-19 restrictions, providing certain clinical significance. The following issues should be addressed.

1. The results showed that RNA sequencing offers a more comprehensive view of the infectome, while DNA sequencing excels in detecting DNA viruses with greater sensitivity. Notable increases in *Mycoplasma pneumoniae*, human respiratory syncytial virus (RSV), and *Haemophilus influenzae* were observed after COVID-19 restrictions were lifted.

Now that the authors have identified this phenomenon, it needs to be explained in the discussion section. Why did this phenomenon occur?

2. Viral-bacterial co-infections were common, with *M. pneumoniae* and RSV being the most prevalent combination.

This phenomenon needs to be explained in the discussion section.

3. By integrating multi sequencing technologies, we uncovered critical trends: a resurgence of virulent pathogens like *Mycoplasma pneumoniae* and RSV after restrictions eased.

Please compare the epidemiology of *Mycoplasma pneumoniae* and RSV before COVID-19 restrictions.

4. Frequent viral-bacterial co-infections linked to severe outcomes.

In this study, including case reports, no serious cases were found. How did the researchers draw this conclusion?

5. Two samples were collected after admission, which sample was used as the indicator of infection? Which time was used as the research indicator in this study?

It was noted that an introduction of each patient after admission was conducted according to Figure 1. But in fact, compared with the microorganisms at the beginning of admission, the microorganisms detected a few days post-admission were affected by the treatment, such as the sensitive or resistant drugs against the microorganisms.

6. To characterize the changes of the respiratory microbiome during disease course of pneumonia, we compared microbial abundances between the first and second time points for each patient (Figure 7). Overall, the abundances of RNA viruses, bacterial pathogens, and fungal pathogens significantly decreased from first to the second time point (all $P < 0.05$, Figure 7A).

How did this phenomenon occur? Was it correlated with the treatment after admission? Was it involved in the discussion? The decline in RNA abundance and bacterial abundance can be explained by the administration of corresponding drugs, but how can the decline in fungal abundance be explained? Are these patients also treated with antifungal drugs?

7. In contrast, DNA viruses showed an increasing trend ($P = 0.081$).

This needs to be explained in the discussion section.

8. While the total commensal bacterial abundance remained stable ($P = 0.45$), genera *Clostridium*, *Lactobacillus*, and *Lactococcus* showed significant increased abundance level (all $P < 0.05$, Figure 7B).

Why did the abundance of these commensal bacteria increase? It should be explained.

9. This suggests the possibility of secondary infections or hospital-acquired infections.

The detection result of the sputum may be related to the hospital environment, and it may

not be an infection. It is recommended to describe it carefully.

10. **Patient 04: Metatranscriptomic results matched hospital tests and revealed additional co-infection with *Candida albicans*.**

This description is not suitable.

11. **By the 31st, he showed improvement, but on the 3rd, while *C. albicans* was cleared and RSV levels decreased, *M. pneumoniae* remained high, and secondary infection with *Staphylococcus epidermidis* emerged, necessitating continued antibiotic treatment (Figure 8B).**

Should the detection of *Staphylococcus epidermidis* be considered as an infection?

12. **On the day of her discharge on 4th, a second sampling was performed. *M. pneumoniae* and *C. albicans* were cleared. However, *H. parainfluenzae* became the dominant pathogen, and RSV and rhinovirus 3 emerged (Figure 8D). This case illustrates the sequential emergence of pathogens during the patient's course.**

The explanation of these case reports is not good enough.

13. **After antibiotic and steroid treatment, the sample on the 8th showed that *P. aeruginosa* was cleared, but *M. pneumoniae* levels remained unchanged, and secondary infection with Human polyomavirus 4 emerged.**

The antibiotic and steroid treatment was not described specifically. Is the antibiotic sensitive to *M. pneumoniae*?

14. **The co-occurrence pattern suggests potential synergistic interactions between *M. pneumoniae* and respiratory viruses in the pathogenesis of pneumonia.**

Why did this phenomenon happen? It should be discussed.

Dear Reviewers:

We sincerely thank you for your time and the comments concerning our manuscript. Those comments are all valuable and very helpful for revising and improving our paper, as well as the important guiding significance to our researches. According to the reviewers' comments, we have made associated modifications to our manuscript to make our results convincing. In this revised version, changes to our manuscript were all highlighted within the document by using red-colored text. Point-by-point responses to the nice associate editor and two nice reviewers are listed below this letter.

Reviewer 1

Major Concerns:

Just as the author mentioned in the Discussion, only samples from two time points were collected, and the sample size was not large enough to explain the dynamic changes of the pathogen comprehensively. Please discuss more whether the author's selection of the sampling time excluded the interference of other factors, in other words, statistically robust.

Response: We sincerely appreciate your valuable comment regarding this issue. As indicated in the Discussion section, we acknowledge the limitations of using only two time points and a relatively small sample size, which may not fully capture the comprehensive dynamics of the pathogen. The selection of these two time points was based on clinical diagnostic support, as these time points correspond to the patient's acute phase and recovery phase, which are considered key stages in the pathogen's lifecycle. We made efforts to standardize sampling conditions and minimize external confounding factors, such as environmental fluctuations. Statistically, we employed Fisher's Exact Test and Wilcoxon Signed-Rank Test for datasets with smaller sample sizes, ensuring that the results provide valuable insights into the dynamic changes of the pathogen during the disease progression.

Minor concerns:

In the discussion section of the manuscript, the author has written many established facts, such as lines 390-393, which seem somewhat redundant. It is suggested to simplify some sentences to make the conclusion appear more explicit.

Response: Thank you for your suggestion. In response to this, we have revised the relevant sentences to streamline the discussion and enhance clarity. We aimed to eliminate unnecessary repetition while maintaining the integrity of the information presented. The revised version now is easier for readers to grasp the main points without being overwhelmed by background details.

1. Line 160-161, please add a citation for the fastp software.

Response: Thanks. We have added the necessary citation for the software in the revised manuscript to properly acknowledge its contribution.

2. Line 233-234, the authors identified "30 pathogen species across 17 microbial families" from all samples. Does this represent all microbes in these samples or a filtered result? If non-pathogenic microorganisms were excluded, what criteria and methods were used to screen for pathogens?

Response: The list of "30 pathogen species across 17 microbial families" represents the filtered result, excluding non-pathogenic microorganisms. First, we excluded microbes with RPM < 1. Then, we referenced several literatures and "List of Pathogenic Microorganisms Infecting Humans" to retain only those pathogens known to infect humans. The criteria and methods used for pathogen screening, along with the relevant references, have been included in the revised manuscript for better clarity. Thank you for your valuable suggestion, which has contributed to enhancing the transparency of our methodology.

3. Line 233-236, We note discrepancies between DNA and RNA sequencing results in Figure 2. For example, several samples detected Human betaherpesvirus 7 via DNA sequencing but not RNA sequencing, whereas *Candida albicans* showed the opposite pattern. Please explain these differences.

Response: Thank you for your insightful question regarding the discrepancies between DNA and RNA sequencing results. The detection of Human betaherpesvirus 7 in DNA sequencing but not in RNA sequencing may suggest that the virus is present in the sample but not actively transcribed at the time of sampling, or that its transcription levels were too low to be detected by RNA sequencing. This aligns with our conclusion that DNA sequencing is more sensitive to microorganisms with low transcription levels. On the other hand, the detection of *Candida albicans* only in RNA sequencing indicates that it was in an actively transcribing state at the time of sampling, but its DNA levels might have been too low in the sample for detection by DNA sequencing. Thank you for your valuable input, which has greatly helped us refine our analysis.

4. Line 245, Figure 4 contains numerous sub-figures that compromise readability. We recommend retaining only representative phylogenetic trees in the main figure and

moving the remainder to supplementary materials.

Response: Thank you for your suggestion. In response, we have integrated the phylogenetic trees from Figures 3 and 4, retaining only the representative trees in the main figures for improved readability. The remaining phylogenetic trees have been moved to Supplementary Figure 2.

5. line 288-289, What does the vertical axis represent? Is it $\log_{10}(\text{RPM}+1)$? Please supplement this information.

Response: Yes, it represents $\log_{10}(\text{RPM}+1)$. We have revised Figure 8, and the y-axis represents $\log_{10}(\text{RPM}+1)$. This information has been added to the figure legend and the manuscript for better clarity.

Reviewer 2

This is a prospective cohort study, which analyzes the NGS results of sputum from children before and after COVID-19 restrictions, providing certain clinical significance.

Response: We sincerely thank the reviewer for their positive evaluation of our study.

Re: Spectrum01450-25R1 (**Dynamics of the Respiratory Infectome in Children with Community-Acquired Pneumonia: Insights from Large and Short Time-Scale Analyses**)

Dear Dr. Yan-Jun Kang:

Thank you for the privilege of reviewing your work. Below you will find my comments, instructions from the Spectrum editorial office, and the reviewer comments.

Revision Guidelines

Sincerely,
Ana Cabrera
Editor
Microbiology Spectrum

Reviewer #1 (Comments for the Author):

The authors have addressed my concerns.

Reviewer #2 (Comments for the Author):

This is a prospective cohort study, which analyzes the NGS results of sputum from children before and after COVID-19

restrictions, providing certain clinical significance.

1. The results showed that RNA sequencing offers a more comprehensive view of the infectome, while DNA sequencing excels in detecting DNA viruses with greater sensitivity. Notable increases in *Mycoplasma pneumoniae*, human respiratory syncytial virus (RSV), and *Haemophilus influenzae* were observed after COVID-19 restrictions were lifted.

Now that the authors have identified this phenomenon, it needs to be explained in the discussion section. Why did this phenomenon occur?

2. Viral-bacterial co-infections were common, with *M. pneumoniae* and RSV being the most prevalent combination.

This phenomenon needs to be explained in the discussion section.

3. By integrating multi sequencing technologies, we uncovered critical trends: a resurgence of virulent pathogens like *Mycoplasma pneumoniae* and RSV after restrictions eased.

Please compare the epidemiology of *Mycoplasma pneumoniae* and RSV before COVID-19 restrictions.

4. Frequent viral-bacterial co-infections linked to severe outcomes.

In this study, including case reports, no serious cases were found. How did the researchers draw this conclusion?

5. Two samples were collected after admission, which sample was used as the indicator of infection? Which time was used as the research indicator in this study?

It was noted that an introduction of each patient after admission was conducted according to Figure 1. But in fact, compared with the microorganisms at the beginning of admission, the microorganisms detected a few days post-admission were affected by the treatment, such as the sensitive or resistant drugs against the microorganisms.

6. To characterize the changes of the respiratory microbiome during disease course of pneumonia, we compared microbial abundances between the first and second time points for each patient (Figure 7). Overall, the abundances of RNA viruses, bacterial pathogens, and fungal pathogens significantly decreased from first to the second time point (all $P < 0.05$, Figure 7A).

How did this phenomenon occur? Was it correlated with the treatment after admission? Was it involved in the discussion? The decline in RNA abundance and bacterial abundance can be explained by the administration of corresponding drugs, but how can the decline in fungal abundance be explained? Are these patients also treated with antifungal drugs?

7. In contrast, DNA viruses showed an increasing trend ($P = 0.081$).

This needs to be explained in the discussion section.

8. While the total commensal bacterial abundance remained stable ($P = 0.45$), genera *Clostridium*, *Lactobacillus*, and *Lactococcus* showed significant increased abundance level (all $P < 0.05$, Figure 7B).

Why did the abundance of these commensal bacteria increase? It should be explained.

9. This suggests the possibility of secondary infections or hospital-acquired infections.

The detection result of the sputum may be related to the hospital environment, and it may not be an infection. It is recommended to describe it carefully.

10. Patient 04: Metatranscriptomic results matched hospital tests and revealed additional co-infection with *Candida albicans*. This description is not suitable.

11. By the 31st, he showed improvement, but on the 3rd, while *C. albicans* was cleared and RSV levels decreased, *M. pneumoniae* remained high, and secondary infection with *Staphylococcus epidermidis* emerged, necessitating continued antibiotic treatment (Figure 8B).

Should the detection of *Staphylococcus epidermidis* be considered as an infection?

12. On the day of her discharge on 4th, a second sampling was performed. *M. pneumoniae* and *C. albicans* were cleared. However, *H. parainfluenzae* became the dominant pathogen, and RSV and rhinovirus 3 emerged (Figure 8D). This case illustrates the sequential emergence of pathogens during the patient's course.

The explanation of these case reports is not good enough.

13. After antibiotic and steroid treatment, the sample on the 8th showed that *P. aeruginosa* was cleared, but *M. pneumoniae* levels remained unchanged, and secondary infection with Human polyomavirus 4 emerged.

The antibiotic and steroid treatment was not described specifically. Is the antibiotic sensitive to *M. pneumoniae*?

14. The co-occurrence pattern suggests potential synergistic interactions between *M. pneumoniae* and respiratory viruses in the pathogenesis of pneumonia.

Why did this phenomenon happen? It should be discussed.

Dear Reviewer,

We are sincerely grateful for the thorough and constructive review of our manuscript. The feedback provided has been invaluable in guiding our revisions, and we have addressed all comments in detail. We believe the manuscript is now substantially improved in both clarity and scientific impact. The major revisions include a comprehensive rewriting of the case studies to better illustrate the clinical utility of our findings, and a significant expansion of the Discussion section to provide robust mechanistic explanations for the observed epidemiological shifts and microbial dynamics. We are confident that the revised manuscript is now a much stronger piece of work. Our detailed point-by-point responses to your comments are provided below.

This is a prospective cohort study, which analyzes the NGS results of sputum from children before and after COVID-19 restrictions, providing certain clinical significance.

1. The results showed that RNA sequencing offers a more comprehensive view of the infectome, while DNA sequencing excels in detecting DNA viruses with greater sensitivity. Notable increases in *Mycoplasma pneumoniae*, human respiratory syncytial virus (RSV), and *Haemophilus influenzae* were observed after COVID-19 restrictions were lifted.

Now that the authors have identified this phenomenon, it needs to be explained in the discussion section. Why did this phenomenon occur?

Response: Thank you for this insightful suggestion. We explain that metatranscriptomics targets actively transcribed RNA, making it highly sensitive for RNA viruses (whose genomes are RNA) and active bacteria. In contrast, metagenomics detects all DNA, making it more suitable for identifying DNA viruses, especially those in a latent or low-transcription state, where their DNA genome is present but RNA transcripts are scarce.

Regarding the resurgence of *M. pneumoniae* and RSV: This is a central finding of our study, and we have now dedicated a significant portion of the Discussion to explaining this critical epidemiological shift. Our primary explanation is centered on the “immunity debt” hypothesis. We propose that the prolonged period of non-pharmaceutical interventions (e.g., masking, social distancing) during the COVID-19 pandemic severely limited the circulation of common respiratory pathogens. This led to a decline in population-level immunity, especially among young children who had minimal prior exposure. Consequently, once restrictions were eased, the large pool of immunologically naive children fueled the intense, widespread, and often off-season outbreaks of pathogens like *M. pneumoniae* and RSV that we documented in our 2023 cohort.

The revised section in the manuscript now explicitly states:

“Following the easing of COVID-19 restrictions, our study documented a sharp resurgence of virulent pathogens, most notably M. pneumoniae and RSV, which re-emerged with off-season peaks and heightened intensity. This trend is particularly striking when contextualized by their pre-pandemic epidemiology. Prior to 2020, extensive research had established both M. pneumoniae and RSV as leading etiological agents of pediatric CAP, each characterized by predictable seasonal circulation patterns (52, 53). The intense resurgence observed in our 2023 cohort strongly aligns with the “immunity debt” hypothesis, a phenomenon reported globally where populations experience increased susceptibility to endemic pathogens following a prolonged period of limited exposure (6, 36). Our findings therefore suggest that the non-pharmaceutical interventions not only suppressed transmission temporarily but also

fundamentally altered the established epidemiological dynamics of these key respiratory pathogens, leading to the widespread outbreaks observed upon the relaxation of control measures (37, 38, 40, 54, 55). Consequently, the simultaneous circulation of a diverse array of respiratory pathogens, coupled with frequent co-infections (23, 26-28) such as the co-infection of RSV and Mycoplasma pneumoniae, likely contributed to the severe and prolonged respiratory disease outbreak among children in China during the autumn-winter season.”

2. Viral-bacterial co-infections were common, with *M. pneumoniae* and RSV being the most prevalent combination.

This phenomenon needs to be explained in the discussion section.

Response: We appreciate you pointing this out. We have added a detailed discussion on the potential synergistic mechanisms between *M. pneumoniae* and RSV in the Discussion section:

*While the precise mechanisms underlying this synergy are not yet fully elucidated, several potential pathways have been proposed based on clinical observations and mechanistic studies of viral-bacterial co-infections. The synergy is likely multifactorial. The key mechanisms appear to involve: (1) virally induced damage to the respiratory epithelium, which enhances bacterial adherence; (2) potential immunosuppression by *M. pneumoniae*, which facilitates viral infection; and (3) a critically dysregulated and amplified host inflammatory response(62-64). While clinical evidence strongly supports the existence of this synergy, further mechanistic studies are essential to fully delineate the molecular interactions and to inform the development of more effective therapeutic strategies for managing severe co-infections”*

3. By integrating multi sequencing technologies, we uncovered critical trends: a resurgence of virulent pathogens like *Mycoplasma pneumoniae* and RSV after restrictions eased.

Please compare the epidemiology of *Mycoplasma pneumoniae* and RSV before COVID-19 restrictions.

Response: Thank you for this important point. As our study cohort was enrolled from 2022 to 2023, we do not have pre-pandemic data for direct comparison. To address this, we have now cited existing literature in the Discussion section to provide the necessary context on the pre-pandemic epidemiology of *M. pneumoniae* and RSV in children (line 478-480). This helps to frame our findings within the broader context of the "immunity debt" phenomenon observed globally.

4. Frequent viral-bacterial co-infections linked to severe outcomes.

In this study, including case reports, no serious cases were found. How did the researchers draw this conclusion?

Response: This is a crucial point, and we thank you for ensuring the precision of our conclusions. You are correct that our study did not include cases with outcomes such as mortality. We have therefore revised our manuscript to tone down the language. We replaced phrases like "severe outcomes" with more accurate descriptions that are supported by our data, such as "prolonged pneumonia," and "complicated disease course" (e.g., in the Abstract, Importance, and Discussion sections).

5. Two samples were collected after admission, which sample was used as the indicator of infection? Which time was used as the research indicator in this study?

It was noted that an introduction of each patient after admission was conducted according to Figure 1. But in fact, compared with the microorganisms at the beginning of admission, the microorganisms detected a few days post-admission were affected by the treatment, such as the sensitive or resistant drugs against the microorganisms.

Response: Thank you for the question regarding our methodology. We have clarified this in the Methods section. Pathogen prevalence was determined based on detection (RPM > 10) in either of the two samples collected during hospitalization. For the longitudinal analysis of microbial dynamics (Figure 7), we explicitly compared the first and second time points. We also acknowledge in the Discussion that the microbial composition of the second sample is inevitably influenced by treatment, and we discuss how this provides valuable information on treatment efficacy and the emergence of other microbes:

Our longitudinal study design, involving two sampling points, offered a unique window into the dynamic nature of the respiratory infectome during hospitalization. We observed that the abundances of most RNA viruses and bacterial pathogens significantly decreased from the first to the second time point (Figure 7A). This trend is plausibly a direct reflection of the efficacy of the antibiotic and antiviral therapies administered to the patients, underscoring the critical importance of the initial sample for accurately identifying the primary etiological agents. Concurrently, the follow-up sample provides invaluable information beyond simple pathogen detection; it serves as a crucial indicator for monitoring treatment response and offers key insights into the emergence of new potential pathogens or shifts in microbial composition that may complicate the clinical course.

6. To characterize the changes of the respiratory microbiome during disease course of pneumonia, we compared microbial abundances between the first and second time points for each patient (Figure 7). Overall, the abundances of RNA viruses, bacterial pathogens, and fungal pathogens significantly decreased from first to the second time point (all $P < 0.05$, Figure 7A).

How did this phenomenon occur? Was it correlated with the treatment after admission? Was it involved in the discussion? The decline in RNA abundance and bacterial abundance can be explained by the administration of corresponding drugs, but how can the decline in fungal abundance be explained? Are these patients also treated with antifungal drugs?

Response: Thank you for this question. We have now explicitly stated in the Discussion that the decline in RNA virus and bacterial abundance is likely a direct result of the antibiotic and antiviral treatments administered to the patients. Regarding the decrease in fungal abundance (*Candida albicans*), as patients did not receive specific antifungal therapy, we propose that this decline is likely due to the recovery of the host's immune system and the restoration of microbial balance following the clearance of the primary pathogens.

The revised section in the manuscript now explicitly states:

*“Interestingly, a similar decline was noted for fungal abundance, particularly for *Candida*. Given that specific antifungal therapy was not systematically administered, this reduction likely follows a*

different mechanism. The decline in Candida is more likely attributable to the restoration of host immune competence as the patients' overall clinical condition improved. Furthermore, it is plausible that in many of these cases, Candida represents a transient colonizer rather than an active pathogen, whose abundance would naturally wane as the primary pathogenic drivers are cleared and the local inflammatory microenvironment is normalized (65, 66). Collectively, these dynamic changes highlight the value of the follow-up sample not just for pathogen detection, but as a crucial indicator for monitoring therapeutic response and understanding the complex ecological shifts within the recovering lung.”

7. In contrast, DNA viruses showed an increasing trend (P = 0.081).

This needs to be explained in the discussion section.

Response: We agree that this trend warrants an explanation. In the revised Discussion, we now hypothesize that the increasing trend of DNA viruses may be due to the reactivation of latent viruses (e.g., herpesviruses, polyomaviruses). This reactivation could be triggered by the acute primary infection and/or the use of steroid treatments, which can modulate the host's immune status.

The revised section in the manuscript now explicitly states:

“We hypothesize that this trend may reflect the reactivation of latent DNA viruses, such as members of the Herpesviridae and Polyomaviridae families detected in our cohort (48, 49). The primary acute infection can induce a state of transient immune dysregulation, which may be further modulated by the administration of corticosteroids—a common treatment in our patient group. This altered immune environment could permit latent viruses to switch from a dormant to a lytic replication cycle, thereby increasing their DNA load at the later time point (50). This observation suggests that the dynamics of the DNA virome could serve as a potential indicator of the host's immune status during recovery from pneumonia (51).”

8. While the total commensal bacterial abundance remained stable (P = 0.45), genera Clostridium, Lactobacillus, and Lactococcus showed significant increased abundance level (all P < 0.05, Figure 7B).

Why did the abundance of these commensal bacteria increase? It should be explained.

Response: Thank you for highlighting this interesting finding. We have added an explanation in the Discussion section. We interpret this as a sign of ecological succession within the respiratory microbiome. As dominant pathogens are cleared by treatment, the vacant ecological niches are repopulated by commensal bacteria, leading to their increased relative abundance. This reflects the dynamic process of microbiome remodeling during recovery.

The revised section in the manuscript now explicitly states:

“Another notable dynamic observed during the course of hospitalization was the significant increase in the relative abundance of certain commensal bacterial genera, namely Clostridium, Lactobacillus, and Lactococcus, while the total commensal abundance remained stable. This phenomenon can be interpreted through the lens of ecological theory, specifically as a case of

microbial succession and niche repopulation. During the acute phase of pneumonia, the respiratory ecosystem is dominated by the high burden of primary pathogens, leading to a state of dysbiosis (67). The administration of antibiotics, while effective in reducing the pathogenic load, creates a profound perturbation that disrupts this established community structure, effectively creating a microbial vacuum or vacant ecological niches (68). In the subsequent recovery phase, the respiratory microbiome begins a process of remodeling. Genera such as Clostridium and Lactococcus, which may be intrinsically more resilient to the administered antibiotics or are opportunistic colonizers originating from the oropharynx or even the gut microbiome, can then proliferate to occupy this newly available ecological space. The rise of these specific genera is therefore not a random occurrence but rather a directed ecological shift (68, 69). This finding is a testament to the resilience and plasticity of the respiratory microbiome and highlights the complex community-level changes that accompany the transition from active disease to clinical recovery. Further exploration of these mechanisms will require in-depth in vitro and in vivo studies, complemented by multi-omics approaches (e.g., transcriptomics, metabolomics, and immunomics) to integrate data from both the host and microbial communities.”

9. This suggests the possibility of secondary infections or hospital-acquired infections.

The detection result of the sputum may be related to the hospital environment, and it may not be an infection. It is recommended to describe it carefully.

Response: We acknowledge that distinguishing true infection from colonization or environmental contamination is a limitation of sequencing-based studies. Following your advice, we have carefully revised the manuscript to use these terms more cautious.

10. Patient 04: Metatranscriptomic results matched hospital tests and revealed additional co-infection with *Candida albicans*.

This description is not suitable.

Comment 10: Patient 04: Metatranscriptomic results matched hospital tests and revealed additional co-infection with *Candida albicans*. This description is not suitable.

Response: Thank you for pointing out the need for a more suitable description and better clinical correlation for this finding. We agree that the original description was insufficient. We have revised the case description for Patient 04 to provide the necessary clinical context that validates the significance of detecting *Candida albicans*. The metatranscriptomic finding is now directly linked to a clinical diagnosis made during the patient's hospital stay.

In the revised manuscript, we have added the following details:

"A dermatology consult confirmed a diagnosis of cutaneous candidiasis, consistent with the high abundance of C. albicans detected in the sputum."

This revision clarifies that the detection of *C. albicans* was not an incidental finding but corresponded to a confirmed clinical condition, thereby justifying its inclusion and importance in the case report.

11. By the 31st, he showed improvement, but on the 3rd, while *C. albicans* was cleared and RSV levels decreased, *M. pneumoniae* remained high, and secondary infection with *Staphylococcus epidermidis* emerged, necessitating continued antibiotic treatment (Figure 8B).

Should the detection of *Staphylococcus epidermidis* be considered as an infection?

Response: We agree that classifying the detection of *Staphylococcus epidermidis*, a common commensal, as a definitive "infection" is a strong claim without further clinical signs of sepsis. To address this, we have revised the manuscript to use more precise and cautious language that accurately reflects the nature of our metatranscriptomic data.

The revised sentence now reads:

"Critically, the microbial landscape shifted, marked by the emergence of transcriptionally active Staphylococcus epidermidis (Figure 8B), a known opportunistic pathogen. This finding, suggesting a potential secondary bacterial complication, supported the decision to continue antibiotic treatment to prevent clinical deterioration."

This change clarifies that our data shows the pathogen was transcriptionally active (and thus not merely a contaminant) and frames it as a "potential complication" rather than a confirmed "infection." This more accurately represents the data while still providing a clear rationale for the clinical decision to continue antibiotics.

12. On the day of her discharge on 4th, a second sampling was performed. *M. pneumoniae* and *C. albicans* were cleared. However, *H. parainfluenzae* became the dominant pathogen, and RSV and rhinovirus 3 emerged (Figure 8D). This case illustrates the sequential emergence of pathogens during the patient's course.

The explanation of these case reports is not good enough.

Response: Thank you for this critical feedback. We acknowledge that the initial case descriptions lacked the necessary clinical detail to create a compelling and clear narrative. We have undertaken a comprehensive revision of all five case studies, integrating detailed clinical timelines, key laboratory markers (e.g., CRP, HBP, PCT, IL-6), and specific treatment information.

For Patient 21, which you highlighted, we have rewritten the description to create a much clearer story. We now detail the patient's initial presentation with high inflammatory markers, the metatranscriptomic finding of *M. pneumoniae*, the subsequent targeted switch in antibiotics to azithromycin, and the resulting clinical improvement. Furthermore, to explain the seemingly paradoxical emergence of new pathogens during recovery, we have added an interpretive sentence.

The revised manuscript now concludes the description for Patient 21 with:

"The emergence of new pathogens during clinical recovery suggests that while the primary pathogenic driver (M. pneumoniae) was controlled, the respiratory microbiome remained in a

volatile state of succession. The newly detected organisms likely represented transient colonizers or subclinical infections that were being effectively managed by the patient's recovering immune system."

We believe this level of detailed, integrated analysis has been applied across all cases and now provides explanations that are scientifically robust and sufficient.

13. After antibiotic and steroid treatment, the sample on the 8th showed that *P. aeruginosa* was cleared, but *M. pneumoniae* levels remained unchanged, and secondary infection with Human polyomavirus 4 emerged.

The antibiotic and steroid treatment was not described specifically. Is the antibiotic sensitive to *M. pneumoniae*?

Response: Thank you for highlighting this crucial point and requesting more specificity. We have revised the description for Patient 53 to include the specific names of the antibiotics used. Most importantly, we now directly address the critical question of antibiotic sensitivity, which powerfully demonstrates the clinical utility of our metatranscriptomic findings.

The revised section in the manuscript now explicitly states:

"This finding was pivotal, as it explained both the severe inflammatory state and the treatment failure. Crucially, M. pneumoniae lacks a cell wall and is intrinsically resistant to the beta-lactam antibiotics the patient had received (both cefuroxime and the subsequent amoxicillin-clavulanate). While the escalated antibiotic regimen could target P. aeruginosa, the persistence of the resistant M. pneumoniae explains the patient's complicated course."

This revision now clearly specifies the treatments and directly explains why *M. pneumoniae* levels did not decrease—the prescribed antibiotics were ineffective against it. This directly answers your question and strengthens the argument that our diagnostic approach provided critical information that could have guided more effective therapy sooner.

14. The co-occurrence pattern suggests potential synergistic interactions between *M. pneumoniae* and respiratory viruses in the pathogenesis of pneumonia.

Why did this phenomenon happen? It should be discussed.

Response: Thank you. This comment is related to Point 2. As mentioned in our response to Point 2, we have significantly expanded the Discussion section to explore the mechanisms behind viral-bacterial synergy. We discuss how prior viral infection can damage the respiratory epithelium and modulate the immune system, thereby creating a favorable environment for *M. pneumoniae* to establish an infection.

Re: Spectrum01450-25R2 (**Dynamics of the Respiratory Infectome in Children with Community-Acquired Pneumonia: Insights from Large and Short Time-Scale Analyses**)

Dear Dr. Yan-Jun Kang:

Thank you for the privilege of reviewing your work. Below you will find my comments, instructions from the Spectrum editorial office, and the reviewer comments.

I am pleased to inform you that your manuscript has been editorially accepted for publication. However, an Ethics Approval statement is missing in the submission, as well as a Data Availability statement. These sections need to be added to Material and Methods before the final decision. Once these are completed, please return your submission so that I can move your paper forward to acceptance.

Revision Guidelines

Sincerely,
Ana Cabrera
Editor
Microbiology Spectrum

Reviewer #1 (Comments for the Author):

The authors have already addressed my concerns.

Reviewer #2 (Comments for the Author):

The authors have adequately addressed the issues raised in the previous review.

Re: Spectrum01450-25R3 (**Dynamics of the Respiratory Infectome in Children with Community-Acquired Pneumonia: Insights from Large and Short Time-Scale Analyses**)

Dear Dr. Yan-Jun Kang:

Your manuscript has been accepted, and I am forwarding it to the ASM production staff for publication. Your paper will first be checked to make sure all elements meet the technical requirements. ASM staff will contact you if anything needs to be revised before copyediting and production can begin. Otherwise, you will be notified when your proofs are ready to be viewed.

Sincerely,
Ana Cabrera
Editor
Microbiology Spectrum